# Integrated approach for characterizing aquifer heterogeneity in alluvial plains

Igor Karlović[1], Mitja Janža[2], Edmundo Placencia-Gómez[2], Tamara Marković[1]

[1] Croatian Geological Survey, Zagreb, 10 000, Croatia
[2] Geological Survey of Slovenia, Ljubljana, 1000, Slovenia

*Correspondence to*: Mitja Janža (mitja.janza@geo-zs.si)

**Abstract.** Alluvial aquifers serve as vital groundwater resources worldwide. Due to their complex heterogeneity, accurate characterization requires the integration of multiple data types. This study presents a systematic framework to address aquifer heterogeneity through hydrofacies analysis, combining borehole data, electrical resistivity tomography (ERT) and stochastic modeling. The approach was tested in the Varaždin aquifer, where geostatistical and stochastic tools were used to simulate the spatial distribution of four hydrofacies: gravel (G), gravel, sandy to clayey (Gsc), sand with gravel, clayey to silty (Sgcs), and clay to silt, sandy (CSs). As the thin and electrically conductive Sgcs-CSs layer limited the ERT resolution below 20 m depth, synthetic models were used to assess their geometry and resistivity magnitudes, estimating a model of these hydrofacies at greater depths. The resulting dimensions of the lens-shaped structures revealed the horizontal extent of the hydrofacies, and were incorporated into horizontal Markov chain models. The 3D Markov chain models were used to generate 10 stochastic realizations of the hydrofacies distribution. Validation identified the representative hydrofacies model for the Varaždin aquifer with a prediction accuracy of 63 %. Results from simulations focused on the Vinokovščak wellfield area show that incorporating ERT-derived lens lengths into the model development slightly improved hydrofacies prediction accuracy by 0.3 to 5.0 %, depending on hydrofacies model grid resolution. The analysis of different grid resolutions demonstrates that increasing model detail beyond the characteristic lens dimensions provided no accuracy improvement, suggesting that the optimal cell size is closely related to the estimated lens lengths. In contrast, coarser grids provide a simplified hydrofacies model, potentially increasing prediction accuracy but losing spatial resolution. This methodology forms a basis for integrating spatial heterogeneity into groundwater models, providing a useful tool for sustainable management in alluvial and similar sedimentary environments.

## 1 Introduction

Alluvial plains, geological formations created by sediment deposition from rivers and streams, often contain complex aquifer systems due to the variability of sedimentary conditions over space and time. This heterogeneity in alluvial aquifers is defined by the spatial distribution of characteristic sediments with distinct hydrogeological properties, i.e., hydrofacies units (Carle, 1999). The accurate characterization of subsurface heterogeneity is essential for successful modeling of groundwater flow and contaminant transport (Zhao and Illman, 2017; Rambourg et al., 2022), controlling the reliability of these models for effective

groundwater management (Guo et al., 2019; Janža, 2009). This accuracy is typically limited by sparse datasets, as simulations are highly dependent on the completeness and accuracy of the data (Gong et al., 2023). To overcome the major challenges in characterizing geological heterogeneity - facies delineation and hydraulic property assignment (Savoy et al., 2017), it is

important to integrate different methods for acquiring relevant datasets for modeling, commonly referred to as "hard" and "soft" data. Hard data are typically obtained through direct observation of outcrops or borehole logs, providing relatively accurate information on the vertical sequence of hydrofacies. However, acquiring these data is expensive and often limited to well-studied locations, resulting in insufficient spatial coverage to capture the horizontal heterogeneity and determine the lateral hydrofacies dimensions.

Consequently, soft data, such as inferred geological informations and qualitative insights from geophysical surveys or conceptual models, are used to provide complementary information on the studied system (Turner, 2021). The use of geophysics has proven to be effective in analyzing aquifer materials (Slater, 2007). In particular, the electrical resistivity tomography (ERT) has been used effectively in sedimentary basins for a variety of applications. Examples include the detection of waste-filled gravel pits (Breg Valjavec et al., 2018), the delineation of landfill leachate plumes (Acworth and Jorstad, 2006),

mapping of buried paleochannels (Green et al., 2005) and floodplain fluvial sediments (Ward et al., 2012), and the identification of spatial heterogeneities to parameterize hydraulic conductivity and permeability reconstructions (De Clercq et al., 2020). Furthermore, previous studies have demonstrated a close relationship between electrical resistivity and hydraulic conductivity in alluvial aquifers (e.g., Mastrocicco et al., 2010; Gernez et al., 2019; Vogelgesang et al., 2020). In recent decades, the modeling and characterization of aquifer heterogeneity, such as structural geometry and hydrofacies tendencies,

have advanced significantly through the use of geostatistical and stochastic methods. In contrast to deterministic models that produce a single, consistent output for a given set of initial conditions, stochastic simulations generate multiple equally probable geostatistical realizations of the subsurface to better capture smaller-scale phenomena (e.g., facies within stratigraphic units) that cannot be adequately modeled using deterministic methods (Hermans and Irving, 2017; Turner, 2021). Well-known stochastic methods for generating realizations of facies distributions include object-based techniques (e.g., Geel and Donselaar,

2007), multiple point statistics (MPS) that rely on training images to capture complex spatial patterns (e.g., Hermans et al., 2015; Gottschalk et al., 2017; Zhou et al., 2018;), and other pixel-based simulation methods such as sequential Gaussian simulation (SGS), sequential indicator simulation (SIS), and transition probability geostatistical simulation (T-PROGS) (e.g., Lee et al., 2007; He et al., 2009; Gong et al., 2023). In addition, several studies have performed comparative analyses to evaluate the ability of different stochastic modeling techniques to characterize heterogeneity (Falivene et al., 2006;

dell'Arciprete et al., 2011; Deveugle et al., 2014). The choice of simulation method depends on both the geological structure and the intended predictions (Scheibe and Murray, 1998).

This study focuses on an alluvial aquifer located in the Varaždin area in northwestern Croatia. As the main water source for approximately 170,000 inhabitants, this aquifer has experienced nitrate contamination in the last decades due to the irresponsible use of organic fertilizers in agriculture and an underdeveloped sewage network (Marković et al., 2022). Thus,

understanding nitrate transport is essential for its sustainable water management. However, previous numerical models simulating groundwater flow and nitrate dynamics in this aquifer were deterministic (Karlović et al., 2022; Šrajbek et al., 2022; Brkić et al., 2021), constrained by hard data and interpolations between boreholes, resulting in a layered representation of the aquifer. In this study, the T-PROGS simulation method (Carle and Fogg, 1996, 1997; Carle et al., 1998; Carle, 1999) was used to generate more realistic 3D representations of subsurface heterogeneity. This method, based on Markov chain models and transition probability matrices as random functions, was chosen for its proven effectiveness in modeling heterogeneity in alluvial environments and other sedimentary environments (Zhang et al., 2006; Frei et al., 2009; Janža, 2009; Engdahl et al., 2010; Koch et al., 2014; Bianchi et al. 2015; Guo et al., 2019). Hydrofacies characterization of alluvial environments based on ERT imaging, supported by geological data from boreholes, has been successfully demonstrated (Berzesio et al., 2007; Mele et al., 2012). This approach can enhance stochastic geological realizations, particularly because the geostatistical characteristics in the T-PROGS method are derived from borehole data, which offer limited geological information in the horizontal direction (He et al., 2014). In the present work, resistivity ERT data is not directly integrated into the stochastic T-PROGS simulations, but is instead used to more accurately estimate hydrogeological features such as the mean horizontal lengths of identified hydrofacies, which are then incorporated as key input parameters in the simulation process.

The main objective of this research is to develop an effective approach that utilizes both hard and soft data to characterize heterogeneity in alluvial aquifers. This comprehensive approach consists of four steps: (1) identification of hydrofacies using borehole data; (2) estimation of the lateral extent of hydrofacies based on ERT measurements; (3) stochastic modeling to generate the spatial distribution of hydrofacies; (4) selection of the most plausible realization of hydrofacies distribution. Other important objectives of this work are to test whether the inclusion ofERT-derived lens lengths into model development improves prediction accuracy of hydrofacies spatial distribution, and to evaluate the influence of grid resolution on prediction accuracy.

## 2 Materials and methods

### 2.1 Site description and hydrofacies characterization

The study was conducted in northwestern Croatia, within the Varaždin alluvial aquifer located in the western part of the Drava River valley (Fig. 1). The aquifer covers an area of about 264 km$^2$, at an altitude ranging from 155 to 200 m above sea level. Detailed descriptions of the geological and hydrogeological settings are available in previous publications (e.g., Karlović et al., 2022; Marković et al., 2022; Brkić et al., 2021). The following text provides a brief overview of the main stratigraphic and hydrogeological characteristics relevant to this study. The aquifer consists mainly of gravel and sand, with varying amounts of silt and clay (Urumović et al., 1990). The changes in the flow patterns and sediment deposition of the Drava River during the Pleistocene and Holocene have resulted in the heterogeneous stratigraphy of the aquifer. According to the layer-based conceptual model, which simplifies the distribution of hydrogeological properties, a low-permeability interlayer divides the

aquifer into two layers (Karlović et al., 2021). The overlying semi-permeable layer of the aquifer is thin or non-existent, indicating a high infiltration potential and an increased vulnerability of groundwater to contamination from surface sources. A very low permeable layer consisting of marl, silt, and clay lies beneath the aquifer.

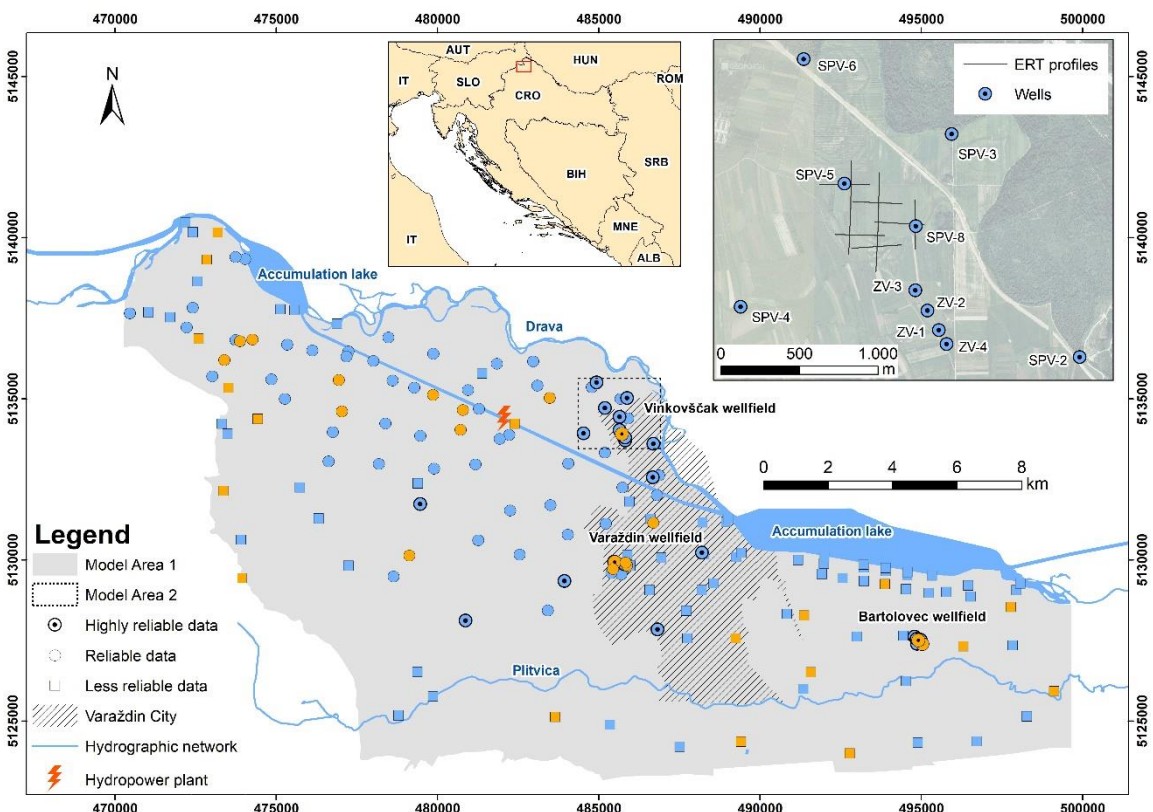

**Figure 1.** Distribution of boreholes and ERT profiles in the study area used for T-PROGS modeling (blue boreholes are used for model development, while orange boreholes represent validation points in Model Area 1). Data reliability classes: highly reliable - original logs and reports available for checking procedures and hydraulic conductivity (K) determination; reliable - original logs and reports available, but lack details on lithology or information for K determination; less reliable - original logs not available, but consistent with nearby reliable data (modified from Ross et al., 2005).

The dataset used in this study to identify hydrofacies consists of 180 boreholes collected from the Croatian Geological Survey database (Fig. 1). Depending on the quality and consistency of the driller's descriptions, the dataset reflects varying levels of reliability, as the borehole logs were collected over decades by multiple investigators. Based on the lithological descriptions from the boreholes, four distinct hydrofacies were defined: gravel (G), gravel, sandy to clayey (Gsc), sand with gravel, clayey to silty (Sgcs), and clay to silt, sandy (CSs) (Table 1). Each lithological unit identified in the borehole logs was assigned to one of these hydrofacies (Fig. 2). All spatial data were organized in ArcGIS software. All maps are presented in the official coordinate system of the Republic of Croatia (HTRS96/TM).

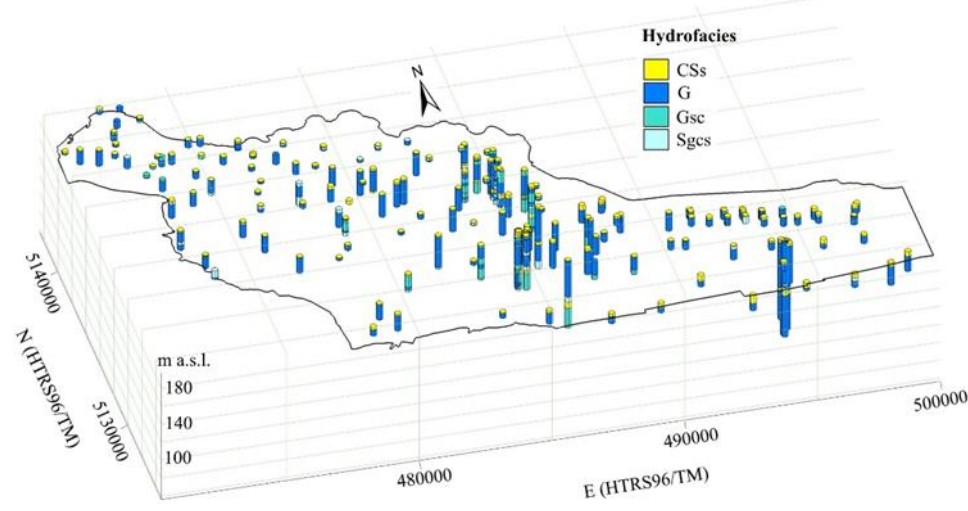

**Figure 2.** Distribution of hydrofacies units in boreholes within the study area.

## 2.2 Application of ERT data to improve characterization of heterogeneity

For modeling heterogeneity, data along the vertical axis are typically obtained from borehole logs at a finer resolution, while horizontal data are limited, with coarser resolutions up to the kilometer-scale, depending on the distance between boreholes. The ERT method was used to better characterize the horizontal extent of hydrofacies. Specifically, a set of 10 ERT profiles was measured in the Vinokovščak wellfield catchment area to estimate the lateral dimensions of hydrofacies, with 5 profiles along (x-axis) and 5 profiles perpendicular to the direction of the Drava River flow (y-axis) (Model Area 2 in Fig. 1). The field measurements were performed in March 2024, using the POLARES 2.0 electrical imaging system. The measurements were taken at a frequency of 20 Hz. Each profile was 315 m long and equipped with 64 electrodes spaced 5 m apart. To collect data, Wenner-Schlumberger (W-S) electrode configuration was used, obtaining a pseudo section of 1250 data points and reaching a maximum depth of investigation of approximately 40 m. Apparent resistivity was inverted using the R2 code, on the ResIPy standalone platform (Blanchy et al., 2020). Lithological information from the SPV-5 and SPV-8 observation wells, located few meters from the nearest ERT profiles VIN-1, VIN-4, and VIN-10 provided hard data to support the interpretation of the resistivity models. In particular, the projection of the boreholes onto the ERT profiles allowed matching the depths and thicknesses of the hydrofacies observed in the boreholes with an iso-resistivity value ($\rho_{hf}$) derived from overlapping the boreholes with contour resistivity maps (as explained below), thereby delineating their resistivity boundaries. This procedure was used to define the resistivity range of the four hydrofacies across the study areausing the three ERT profiles (VIN-1, VIN-4, and VIN-10) as references. Then, by simple delineation of the $\rho_{hf}$ boundary values using the contour resistivity maps, the lateral and vertical extents of each hydrofacies in the other ERT profiles were determined. However, artifacts and a high degree

of uncertainty in the inverted ERT images may occur due to the complexity of structural geometries (e.g., lenses), emplacement, depth, thickness, and resistivity contrast among different geologic materials, as well as limitations inherent to the inversion approaches associated with parameters settings. In the study area, ERT sensitivity was affected by very low resistivity values at approximately 20 m depth, resulting in a loss of resolution and limited depth of investigation, thus preventing to delineate with certainty the extent of $\rho_{hf}$ for such conductive material. To improve our interpretation of ERT dataand better estimate geometric characteristics of hydrofacies below 20 m depth, a series of ERT measurement simulations were performed using synthetic models. These models tested different possible structures such as a continuous, layered lens or discrete, smaller lens-shaped conductive material. The simulations replicated the electrode array (W-S) and sequence from the field. The thickness (vertical extent) of hydrofacies in the synthetic models was constrained using observations from nearby wells (SPV-5 and SPV-8), while the lateral extent of hydrofacies above 20 m depth was approximated based on length estimates from the field-data based model.

## 2.3 Modeling the spatial distribution of the hydrofacies

The spatial distribution of the hydrofacies at the site was modeled using a combination of geostatistical and stochastic methods using T-PROGS software (Carle, 1999) within the Groundwater Modeling System 10.4 platform (Aquaveo, 2018). This approach uses transition probabilities derived from boreholes and a three-dimensional Markov chain model to integrate conceptual geological information, forming a realistic model of subsurface heterogeneity (Carle and Fogg 1996, 1997; Carle et al. 1998). In this study, borehole depth intervals were classified into four hydrofacies based on the borehole log descriptions (Table 1). The hydrofacies models were constructed at different scales, regional and local. The regional model, referred to as Model Area 1 (MA1), represents the entire aquifer, while the local model, referred to as Model Area 2 (MA2), focuses on the Vinokovščak wellfield (Fig. 1).

### 2.3.1 Model Area 1

Of the 180 boreholes in MA1, 80 % were used for model development (n=144), while the remaining 20 % were used for validation (n=36). Transition probability curves were calculated using a lag interval of 0.3 m, which is less than the minimum hydrofacies thickness in 144 boreholes. These curves were used to construct a Markov chain models in the vertical (z) direction. The maximum entropy approach was used to fit the vertical Markov chain models to the measured transition probabilities. The maximum entropy factors represent the ratio between the observed and maximum entropy transition rates. A factor of 1 indicates a random distribution of hydrofacies, depending only on their proportions. Values greater than 1 indicate transitions between hydrofacies that are more frequent than random and vice versa. The probabilistic constraints of the Markov chain model eliminate the need to specify transition rates for the background category, as they are automatically adjusted to balance the equations (Carle, 1999). Based on the ERT interpretation, the Gsc hydrofacies was selected as the background material, filling areas not occupied by other hydrofacies. Mean lengths and widths for non-background hydrofacies were assigned based on ERT profile interpretations (Table 1). The x, y, and z Markov chain models were then combined into a 3-D Markov chain

model that provided input to a conditional simulation that resulted in multiple (n=10), equally probable 3-D realizations of the spatial distribution of the hydrofacies. The grid was configured as 100x100x100 cells in the x, y and z directions, resulting in 1,000,000 cells. The selection of validation boreholes (n=36) considered their spatial distribution as well as their depth and was performed in four steps: i) boreholes were grouped into three depth categories; ii) borehole proportions were calculated for each depth category; iii) boreholes were randomly selected within each depth category; iv) validation boreholes were compiled proportionately from each depth category. Finally, 10 stochastic 3-D realizations of the hydrofacies spatial distribution were compared with the corresponding borehole data in each cell at 1 m vertical resolution. This validation process allowed the identification of the most plausible realization of the spatial distribution of the hydrofacies, with accuracy expressed as the percentage of correct predictions.

### 2.3.2 Model Area 2

The hydrofacies models in MA2 were constructed using the same procedure as in MA1, based on data from 10 highly reliable boreholes in the Vinokovščak wellfield. The model depth was limited to the top 20 m to manage the computational load and to test: i) whether the inclusion of soft data, specifically ERT-derived lens lengths, improves model prediction accuracy compared to the model developed using only borehole data, and ii) the effect of grid resolution on prediction accuracy. Accordingly, mean lens lengths for non-background hydrofacies, derived from ERT profile interpretations, were adjusted to include only lenses within the first 20 m (Table 1). A leave-one-out validation procedure was applied across 10 boreholes, checking 10 realizations for both ERT-derived and default lens lengths in T-PROGS (i.e., 10 times the hydrofacies thickness), resulting in 200 simulations per grid resolution. In total, 1200 simulations were conducted in MA2, using grid resolutions of 10x10x1 m, 20x20x1 m, 40x40x1 m, 60x60x1 m, 80x80x1 m, and 100x100x1 m.

### 3 Results and discussion

### 3.1 Implementation of ERT data to improve characterization of hydrofacies

The entire ensemble of the 10 ERT profiles resulted in a pseudo 3D resistivity model that allowed visualization of the subsurface electrical resistivity distribution up to about 40 m depth throughout the study area (Fig. 3). The model shows a broad range of resistivity values from 60 to 4677 ohm·m from the surface to about 20 m depth, reflecting the degree of heterogeneity characteristic for alluvial environments. However, at greater depths, from 20 to 40 meters, we observe low resistivity values ($\leq$ 100 ohm·m), which limit the depth of investigation and reduce the resolution of ERT measurements. At shallow depths, a clear transition from a low to intermediate resistivity zone in the north to a high resistivity zone in the south is observed, suggesting a progression from fine-medium size to coarser material, consistent with the lithological information provided from boreholes. However, a very high resistivity anomaly in the western part of VIN-1 profile suggests the presence of coarser material in this area. High resistivity anomalies linked to coarse materials mostly appear as elongated, lens-shaped bodies with a flat top surface. Their thickness varies over a depth range from 5 m to 20 m, although they are often distinguished

near the surface, resulting in a well-defined lateral resistivity contrast with surrounding intermediate and low resistivity values within the first 5 m, particularly towards the southern part of the study area. Intermediate resistivity values are consistently observed across all profiles, representing the background material within which the high and low resistivities are embedded.

There is a strong continuity of resistivity in the intersections between profiles, providing a good 3D approximation of the shape and extent of high and low resistivity anomalies at the scale of the study area, i.e., a rough estimate of the lateral and vertical extent of hydrofacies. From a depth of about 20 m, the observed low resistivity anomaly coincides with the presence of thin layers of fine material observed in the boreholes, i.e., hydrofacies CSs and Sgcs. However, the bottom of this layer is not resolved in the ERT imaging, preventing to delineate the lateral and vertical extent of these hydrofacies by the electrical signature, which is explained by the high conductivity values of these materials. Consequently, all the materials underneath is masked with underestimated resistivity values. This interpretation is corroborated by borehole data, which consistently show that Gsc hydrofacies occurs below the CSs and Sgcs units.

The pseudo 3D model based on ERT data was not sufficient to obtain detailed lateral and vertical extent of hydrofacies below 20 m depth, which presents input information for stochastic modelling and predicting spatial distribution of hydrofacies. Therefore, we modified our strategy to interpret the ERT inverted data for each profile by generating contour maps of iso-resistivity values (0.1 ohm·m resolution, log scale) through kriging interpolation over a refined grid mesh of four cells between electrodes. As a result, we obtained ERT images for each profile, as illustrated by the VIN-1 example in Fig. 4, which demonstrates our interpretation approach and serves as the basis for our final analysis. By correlating lithological information from boreholes with iso-resistivity contour maps, we determined the characteristic resistivity values ($\rho_{hf}$) for each hydrofacies at every vertical lithological change (Table 1). For example, the depth boundaries of gravel (G) in boreholes SPV-5 and SPV-8 corresponded to the 500 ohm·m iso-resistivity contour in profiles VIN-1, VIN-4, and VIN-10, establishing this value as the lower resistivity threshold for hydrofacies G. Following the same procedure, intermediate-high resistivity values (200–500 ohm·m) are associated with Gsc, low-intermediate resistivity range (100–200 ohm·m) is linked to Sgcs, and low resistivity values (< 100 ohm·m) correspond to CSs. We note that the $\rho_{hf}$ boundary values for the CSs and Sgcs materials were determined by matching their depth intervals (20-23 m and 0-3 m in SPV-5, and 25-29 m and 0-3.4 m in SPV-8), with the 100 ohm·m and 150 ohm·m in the nearby VIN-1, VIN-4, and VIN-10 profiles, respectively.

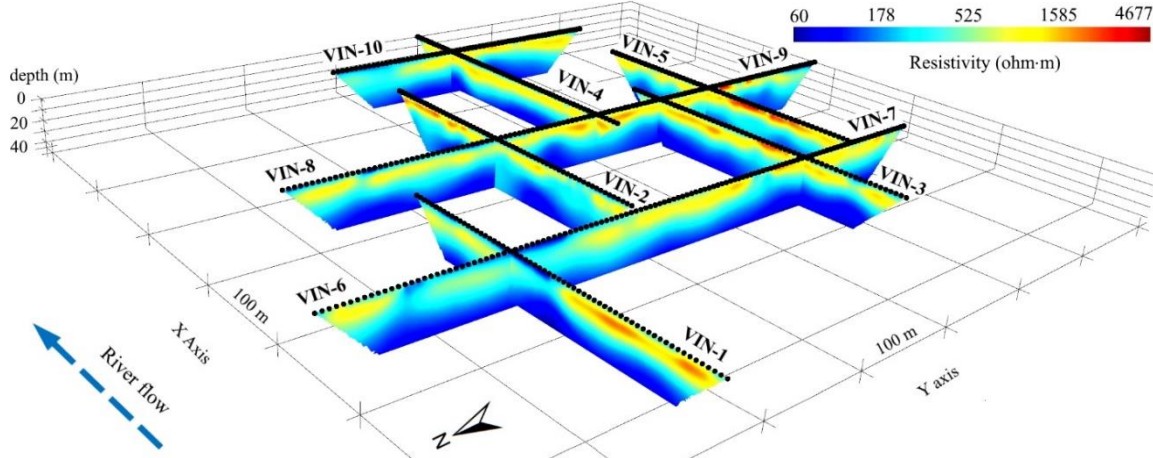

**Figure 3.** Pseudo 3D ERT model for the 10 profiles measured in the Vinokovščak wellfield.

The refined field data-based geoelectrical model for VIN-1 (Fig. 4a) indicates that the CSs material, found at 0–2 m and 20.7–22 m depth in borehole SPV-5, is delineated by the continuous and undulated conductivity line contour at $\rho_{hf}$ = 100 and 125 ohm·m, respectively. However, the line contour at $\rho_{hf}$ = 80 ohm·m suggests a separated conductivity anomalies at 60 m and

160 m distance, with the latter located very close to the projection distance of SPV-5. Another well-defined and separated conductive anomaly is observed from 220 m to 280 m distance with $\rho_{hf}$ = 125-160 ohm·m (Fig. 4a).

These results led us to conduct synthetic models to assess whether the lateral extent of CSs and Sgcs materials in the study area could be better characterized as either a single continuous layer or discontinuous lenses. Given that the contour lines at $\rho_{hf}$ = 80 ohm·m and $\rho_{hf}$ = 160 ohm·m in VIN-1 suggest separated continuous lenses below 20 m depth, we focused our synthetic

modelling on two co-existing lenses with both CSs and Sgcs materials emplaced together (Fig.4c), systematically varying the length, separation distance and resistivity contrast (theoretical values) relative to the other hydrofacies. The proposed lens-shaped geometry is supported by borehole data, with the incomplete presence of this layer indicating its discontinuous nature across the study area.

After evaluating over 20 geological scenarios, including a single continuous layer at 20 m depth, the model in Fig. 4c

demonstrated the closest match (the geoelectrical model in Fig. 4b) to the field data-based model shown in Fig. 4a. The ResIPy software based on R2 was very useful to draw the rounded geometry of hydrofacies in synthetic models, using the same triangular mesh as for the inversion of field data. The acceptance criteria of synthetic simulations were based on the similarities in the shape of the conductive anomaly appearing at 20 m depth in the synthetic model and the equivalent anomaly in the field-data inversion model. In the synthetic models, the geometries of the materials and their resistivity values within the first 20 m of depth were set according to the ranges suggested by the field ERT model. The optimal synthetic model configuration was

achieved by reducing CSs and Sgcs resistivities at 20 m depth to 20 ohm·m, which are values more representative of clay-silt

materials. This implies that estimated resistivity values from field data inversion are likely overestimated for these hydrofacies below 20 m depth.

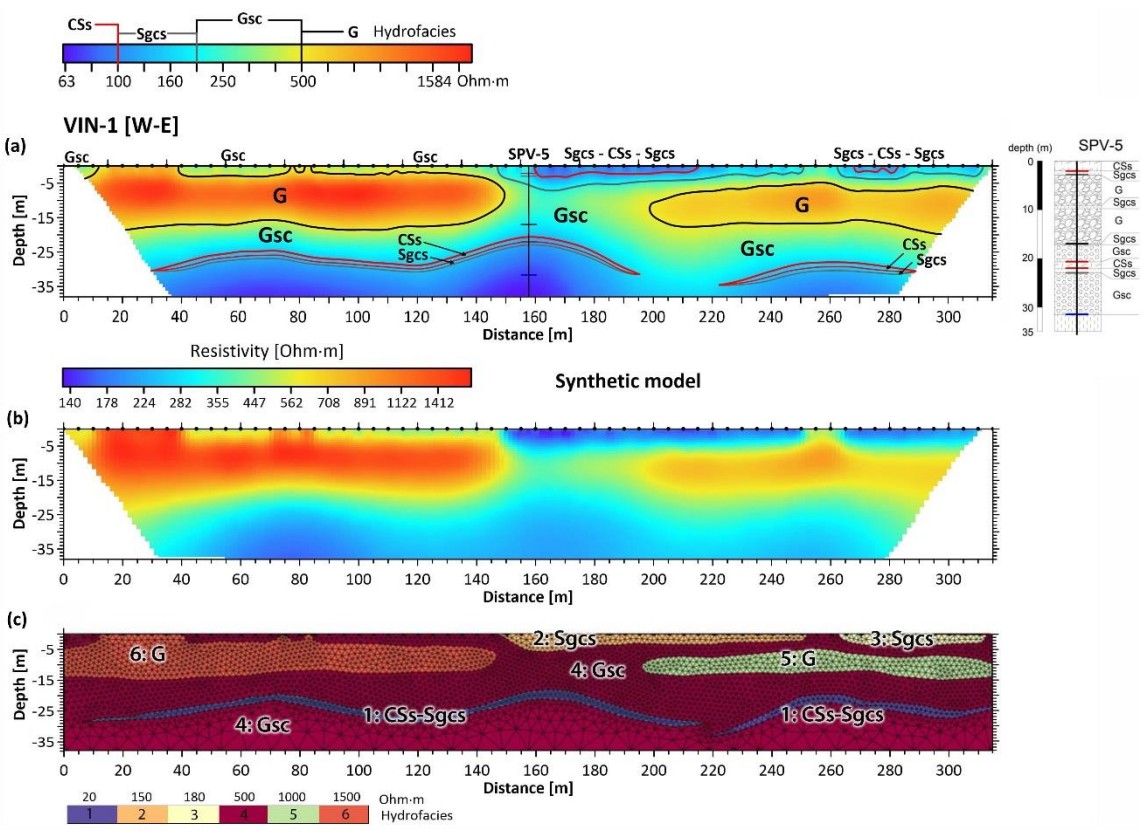


**Figure 4.** ERT profile VIN-1 (RMS misfit = 1.23 %) showing the distribution of hydrofacies, built from: **(a)** joint interpretation of field ERT imaging and lithological information observed in borehole SPV-5; **(b)** coupled with ERT imaging results (RMS misfit = 1.01%) from a synthetic model; **(c)** based on lens-shaped geometries of hydrofacies and associated resistivity values.

The combined ERT results at VIN-1 show a heterogeneous subsurface configured by resistive and conductive lens-shaped
structures. Within the first 20 m, the ERT methodology provides an excellent characterization of hydrofacies, as confirmed by synthetic modeling. Although the resistivity values are overestimated, the lateral and vertical extent of the hydrofacies are well resolved and consistent with the field ERT results. Specifically, the first 20 m of depth in VIN-1 consist of two highly resistive hydrofacies G lenses with average values of 1500 ohm·m and 800 ohm·m, embedded in less resistive hydrofacies Gsc (Fig. 4a). It is important to note a discrepancy between hydrofacies G and the associated $\rho_{hf}$ values at the projected position of
borehole SPV-5, suggesting that the borehole does not intersect the VIN-1 profile at its actual location. However, approximately 15 m from the projected borehole, the thickness of hydrofacies G in SPV-5 aligns perfectly with the high-resistivity anomaly towards the western part of the profile, further highlighting the lateral heterogeneity at the site. Moreover, given that the ERT profile endpoints were recorded with a pocket GPS, potential inaccuracies in horizontal positioning may

have contributed to this discrepancy. Below 20 m depth, gradual decrease in resistivity is observed, from 200 ohm·m to 63

ohm·m at the maximum depth. The iso-resistivity contour lines, which outline the shape and extent of the conductive anomalies, reveal the presence of two lens-shaped conductive bodies in the VIN-1 profile, as suggested by ERT results from synthetic modeling (Fig. 4b-c). Hydrofacies CSs and Sgcs, observed in borehole SPV-5 between 20.7 m and 23 m, align well with the top of the conductive anomaly between the iso-resistivity lines at 160 ohm·m (20 m depth) and 100 ohm·m (25 m depth). Using the same procedure, the conductive anomaly at the end of profile VIN-1 corresponds to a CSs-Sgcs lens, bounded

by the iso-resistivity values of 160 ohm·m and 125 ohm·m between 28 m and 32 m depth. The same approach was systematically applied across all 10 ERT profiles, allowing comprehensive estimation of mean hydrofacies dimensions in both horizontal directions, derived from all identified lenses (Table 1). The synthetic modeling results improved the procedure for constructing hydrofacies models using ERT data by suggesting reliable estimates of hydrofacies dimensions below 20 m depth, which serve as critical input parameters for the T-PROGS model.

**Table 1.** Attributes of the hydrofacies

| Hydrofacies | G | Gsc | Sgcs | CSs |
|---|---|---|---|---|
| Common descriptions | Gravel, sandy, medium-coarse grained | Gravel, sandy-clayey, fine-medium grained | Sand with gravel, clayey-silty | Clay to silt, sandy |
| Electrical resistivity (ohm·m) | >500 | 200–500 | 100–200 | < 100 |
| Model area 1 | | | | |
| Mean thickness (m)[a] | 11.81 | 4.42 | 1.97 | 1.54 |
| Mean length (m)[b] (n[*]) | 230 (6) | background material | 77 (20) | 91 (14) |
| Mean width (m)[c] (n[*]) | 213 (6) | background material | 93 (17) | 111 (12) |
| Volumetric proportion (-) | 0.73 | 0.11 | 0.07 | 0.09 |
| Model area 2 | | | | |
| Mean thickness (m)[a] | 7.45 | 5.62 | 1.89 | 1.63 |
| Mean length (m)[b] (n[*]) | 230 (6) | background material | 34 (11) | 20 (5) |
| Mean width (m)[c] (n[*]) | 213 (6) | background material | 33 (10) | 24 (5) |
| Volumetric proportion (-) | 0.50 | 0.26 | 0.16 | 0.08 |

a: Mean thickness determined from diagonal entries in the vertical transition rate matrix (Carle, 1999)

b: Mean length estimated according to ERT profiles in the direction of the Drava River flow (x-axis)

c: Mean width estimated according to ERT profiles perpendicular to the Drava River flow (y-axis)

*: number of lenses (n) analyzed for each parameter

 **3.2 Transition probability geostatistical simulation**

**3.2.1 Model Area 1**

The Markov chain model identified the vertical tendencies of the hydrofacies in the borehole data (Fig. 5a). The volumetric proportion and mean thickness of each hydrofacies are shown in Table 1. The mean thickness is determined along the diagonal elements of the matrix, representing auto-transitions. Hydrofacies G is the thickest, followed by GSc, Sgcs, and CSs. Since GSc is assigned as a background hydrofacies, its transition rates are computed in relation to the transition rates of other hydrofacies. The entropy factors (EF) observed between hydrofacies pairs show similar, near-random, or below random vertical tendencies (Fig. 5a). The only significant difference is between CSs and Sgcs, with a preference for CSs to transition into Sgcs (EF 1.65) rather than vice versa (EF 0.59). Hydrofacies Sgcs tends to occur above G (EF 1.26), although less frequently than the reverse sequence (EF 1.53). The occurrence of G over CSs (EF 0.89) and the reverse sequence (EF 0.35) are less probable than random. The lack of consistent vertical transition patterns between hydrofacies suggests that their relative proportions play an important role in determining their spatial distribution.

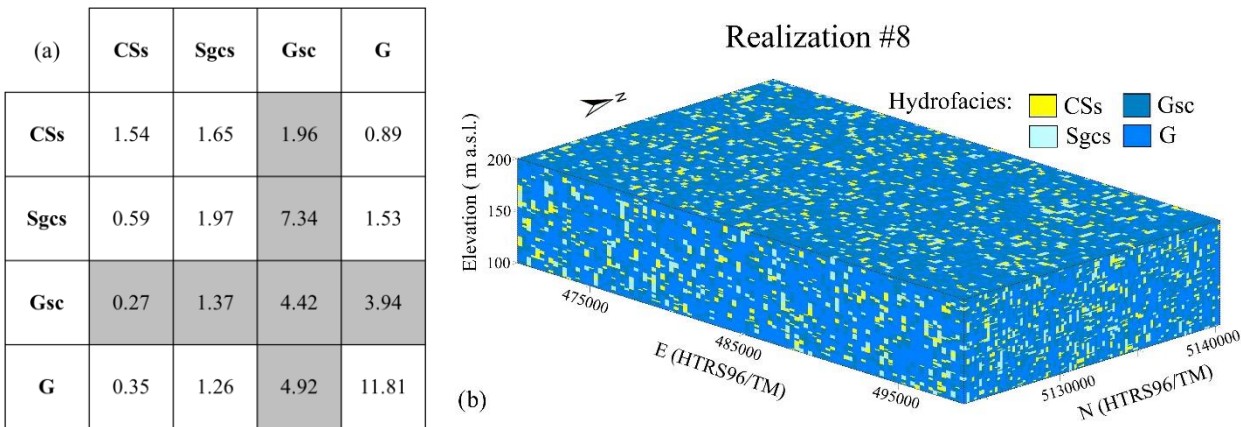

**Figure 5.** T-PROGS results in MA1: **(a)** Entropy factors in the vertical direction generated by a Markov chain model from borehole logs - diagonal boxes represent unobservable auto-transitions, gray boxes display values computed for the background material;**(b)** the representative stochastic hydrofacies model of the Varaždin aquifer (vertical exaggeration is 50-fold).

The integration of ERT-derived lens lengths into the model facilitated the development of horizontal Markov chain models. Lateral continuity of hydrofacies is observed, with similar mean lengths in both horizontal directions, ranging from 18 times (G in the y-direction) to 72 times (CSs in the y-direction) greater than the vertical thickness recorded in the borehole data, depending on the hydrofacies (Table 1). The 3D Markov chain models were used to generate 10 stochastic realizations of the hydrofacies distribution. The validation results identified the most plausible realization of the hydrofacies distribution, i.e., the representative hydrofacies model of the Varaždin aquifer (Fig. 5b). The percentage of correct predictions across 10 realizations ranges from 53 to 63 % (mean 57.3 %, standard deviation 2.5 %), with realization 8 (R8) being the most accurate (Fig. 6a). These values are consistent with those observed in heterogeneous environments, as reported by Bianchi et al. (2015), where

prediction accuracy ranges from 47 to 57 %, and by He et al. (2014), with values between 33 and 77 %, depending on the inclusion of soft data in model development. Furthermore, mismatches between hydrofacies G and Gsc contribute to 16 % of discrepancies. This difference has no significant impact on the assignment of hydraulic conductivities for the groundwater flow model, as both hydrofacies are highly conductive. Two validation boreholes in R8 achieved 100 % prediction accuracy (Fig. 6b), both located in the northwestern part of the study area. Apart from this, no clear trend emerges, with lower prediction accuracy occurring in central areas, as well as near the western and southern edge of the aquifer.

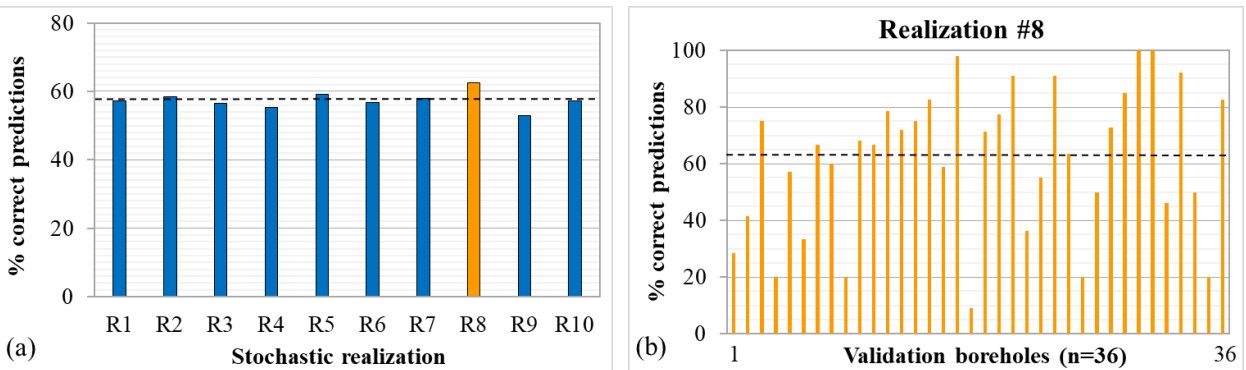

**Figure 6.** Validation test results: **(a)** percentage of correct predictions in 10 stochastic realizations; **(b)** percentage of correct predictions in boreholes for representative realization R8 (dashed lines indicate the mean percentage of correct predictions).

A detailed R8 analysis revealed that the model reproduces spatial distribution of the G hydrofacies most accurately, with an 86 % match between the model and the validation boreholes. The matches of the other hydrofacies are considerably lower, with 18 %, 15 %, 13 % for CSs, Sgcs, and Gsc, respectively. The significantly higher accuracy in the case of hydrofacies G can be attributed to its high volumetric proportion (73 %) and consequently its high reproducibility when using a coarse grid (which exceeds the dimensions of the other hydrofacies). Estimation of the volumetric proportions of hydrofacies is based on borehole data of varying reliability. The purpose of many boreholes used in the study, classified as either reliable or less reliable, was originally to determine aquifer boundaries and not to delineate intervals of deposits with different sand and gravel ratios, such as Sgcs and Gsc.

### 3.2.2 Model Area 2

The simulations in MA2 were performed to evaluate whether incorporating ERT-derived lens lengths improves model prediction accuracy, compared to models developed using only borehole data and default lens lengths. To ensure that the results are not influenced by data reliability, the simulations were conducted in the Vinokovščak wellfield area, using only highly reliable borehole data available in this area. In addition, the model depth was limited to the upper 20 m, corresponding to the depth interval where the ERT methodology provided high-quality characterization of hydrofacies, thus avoiding the effects of low resistivity anomalies and any potential ambiguities from synthetic modeling at greater depths. The limited area

of MA2 also allowed testing the impact of different grid resolutions on model prediction accuracy, as the T-PROGS software supports simulations with up to 3.5 million cells (XMS Wiki, 2025). The Markov chain model identified the vertical tendencies of hydrofacies in 10 boreholes within MA2 (Fig. 7a). As in MA1, the entropy factors in MA2 indicate an absence of vertical patterns, pointing to the relative proportions of hydrofacies as the main driver of their spatial distribution.

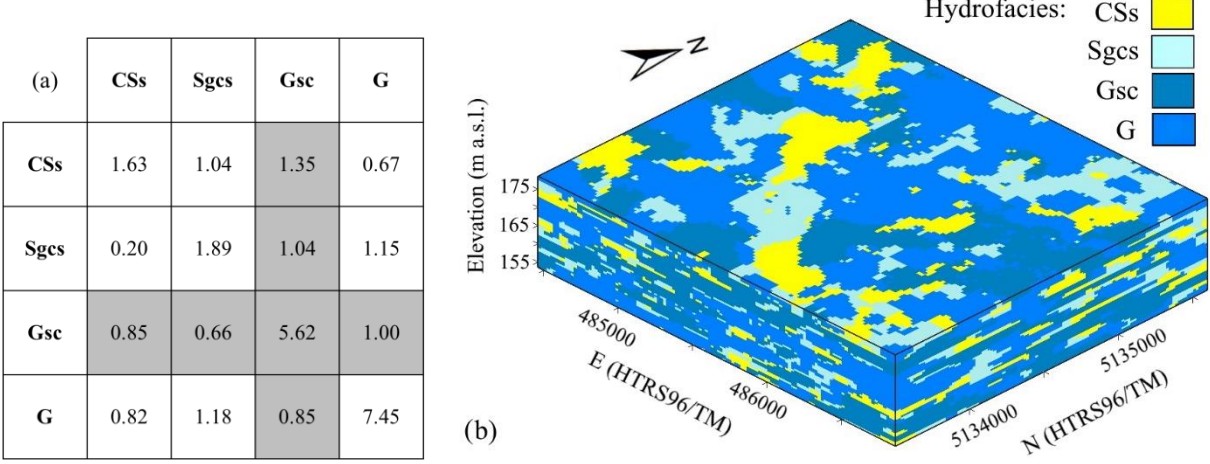

| (a) | CSs | Sgcs | Gsc | G |
|-----|-----|------|-----|---|
| CSs | 1.63 | 1.04 | 1.35 | 0.67 |
| Sgcs | 0.20 | 1.89 | 1.04 | 1.15 |
| Gsc | 0.85 | 0.66 | 5.62 | 1.00 |
| G | 0.82 | 1.18 | 0.85 | 7.45 |

**Figure 7.** T-PROGS results in MA2: **(a)** Entropy factors in the vertical direction generated by a Markov chain model from borehole logs; **(b)** one of the geostatistical realizations of the spatial distribution of hydrofacies in the Vinokovščak wellfield area, constructed using ERT-derived lens lengths with a grid resolution of 20x20x1 m (vertical exaggeration is 20-fold).

After combining vertical and horizontal Markov chain models, developed using both ERT-derived (Table 1) and default lens
lengths (i.e., 10 times the hydrofacies thickness), the 3D Markov chain models were used to generate 10 stochastic realizations of hydrofacies distribution by leaving one borehole out of each simulation. This process was repeated for all 10 boreholes, resulting in 200 simulations per grid resolution. Figure 8 displays a comparison of the MA2 simulation results, showing the prediction accuracy for horizontal grid resolutions of 10x10 m, 20x20 m, 40x40 m, 60x60 m, 80x80 m, and 100x100 m.

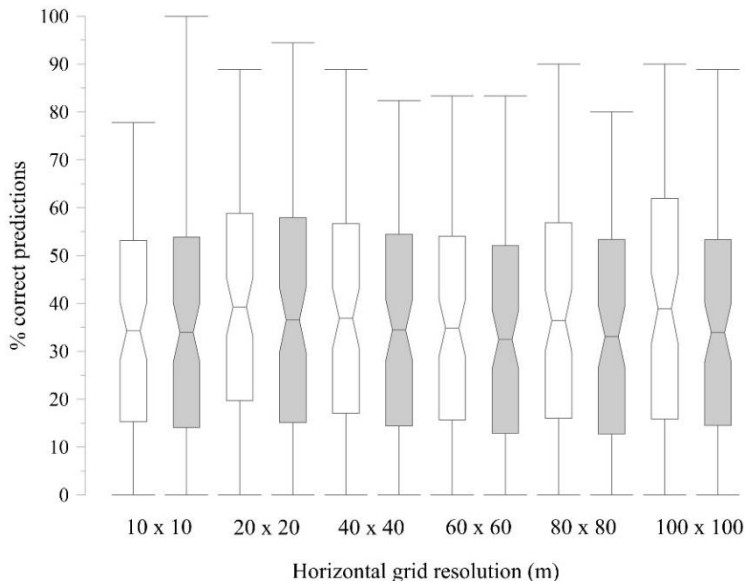

**Figure 8.** Comparison of prediction accuracy of models in the Vinokovščak area based on ERT-derived lens lengths (white box-whisker plots) and default lens lengths (gray plots) at different grid resolutions. Note: The edges of the box represent the standard deviation, the whiskers represent the minimum and maximum, and the central line represents the mean of the data.

The mean values of the correct predictions for simulations based on ERT-derived lens lengths are consistently higher for all grid resolutions compared to models using default data. However, the differences are not significant and range from 0.3 to 5.0

% depending on the grid resolution. The standard deviations indicate that models based on both datasets exhibit wide variability across all grid resolutions. In addition, the min-max ranges are relatively consistent, suggesting that both models handle prediction extremes similarly, regardless of grid resolution. The similar prediction outcomes between the two approaches may be attributed to the comparable lens lengths derived from ERT and default data, with ERT-to-default length ratios varying by hydrofacies: 2.9–3.1 for G, 1.8 for Sgcs, and 1.2–1.5 for CSs (calculated according to the data in Table 1). The relationship

between model prediction accuracy and grid resolution reveals a distinct pattern that requires further exploration. To better understand this pattern, it is important to highlight that the ERT-derived lens lengths used in the MA2 simulations are 20 and 24 m for CSs, and 33 and 34 m for Sgcs (Table 1). The prediction accuracy of the models using 10x10 grid is the lowest, despite its high spatial resolution. This suggests that increasing model detail beyond the characteristic lens dimensions produced subdivisions that did not enhance model performance, reducing the potential benefits of using fine resolution grids.

In contrast, the 20x20 m horizontal cell size closely matches the lens lengths and better captures the spatial patterns of the ERT data. It seems that the 20x20 grid resolution resolves the lens features (Fig. 7b), avoids excessive segmentation and provides the most reliable simulation results, and is therefore the optimal configuration. As the grid resolution increases to 40x40 and 60x60, a decrease in mean prediction accuracy is observed for both data sets. The coarser grid resolution leads to a decrease in the spatial representation of hydrofacies, as larger grid cells cannot adequately resolve the lens lengths. Interestingly, the

trend changes for the 80x80 and 100x100 grid resolutions. Contrary to intuitive expectations, these grids show better prediction accuracy than the 60x60 grid resolution. A possible explanation for this is that larger grid cells provide a more homogeneous representation that better matches validation borehole data. This smoothing effect may help to mitigate inaccuracies caused by incomplete representation of lens lengths, resulting in better performance despite compromising the spatial resolution of the hydrofacies representation.

**4 Summary and conclusions**

The characterization of aquifer heterogeneity in alluvial plains requires the integration of geological, geophysical, geostatistical, and modeling tools. Advancing these methods and improving data integration is crucial for better understanding and management of these vital groundwater resources. The presented hydrofacies model is the first hydrogeological representation of the studied aquifer developed using geostatistical processing and stochastic modeling. Its advantages lie in the transparency and reproducibility of scientifically based procedures. The hydrofacies distribution can be used as a basis for defining hydraulic conductivity fields, a critical input for groundwater flow and contaminant transport modeling, which will support future aquifer management in the study area. The four-step approach summarized below is straightforward, and adaptable to other alluvial or similar sedimentological environments.

*(1) Identification of hydrofacies using borehole data.*

The ability to model aquifer heterogeneity depends on the quality and spatial distribution of hard data, such as boreholes. Due to the varying quality and consistency of borehole logs, the dataset may reflect different levels of reliability, as borehole logs are often compiled over decades by different investigators. Interpreting borehole descriptions and classifying them into hydrofacies is challenging, as well as subjective. It is recommended to use no more than five hydrofacies, as additional categories rarely justify the increased detail and time required (XMS Wiki, 2025).

*(2) Delineation of lateral extent of hydrofacies using ERT.*

Complex heterogeneous environments, such as alluvial aquifers, can be difficult to characterize using simple resistivity data analysis. Therefore, more robust approaches such as the one proposed in this study are needed. A joint interpretation of ERT, borehole data, and synthetic ERT modeling resulted in a more reliable delineation of the hydrofacies below 20 m. This approach helped to overcome some of the limitations of the ERT method, in particular the presence of a thin, electrically conductive layer at 20 m depth. This layer prevented the current from penetrating deeper, which affected the ERT resolution and limited its ability to accurately resolve lens lengths below this depth. While synthetic modeling addressed this issue, other techniques, such as induced polarization and GPR, should be considered in future research to strengthen hydrofacies characterization.

*(3) Stochastic modeling to define the spatial distribution of hydrofacies.*

Developing vertical Markov chain models from borehole data requires accurate fitting of Markov chain curves to measured transition probabilities and assignment of lag distances that reflect all hydrofacies occurrences in boreholes. In addition to the maximum entropy approach used in this study, the modeler can choose between four alternative fitting approaches (Carle, 1999). The entropy factor analysis indicates a lack of consistent vertical transition patterns between hydrofacies, highlighting the importance of relative proportions in shaping their spatial distribution. Hydrofacies lengths from the ERT interpretation

showed dominant horizontal continuity relative to thickness and supported the development of horizontal Markov chain models. T-PROGS demonstrated robust integration of borehole and ERT data into a geologically meaningful 3D representation of the subsurface heterogeneity. Although this study used T-PROGS software, other stochastic approaches that integrate borehole and geophysical data for hydrofacies distribution modeling can be considered.

*(4) Selection of the representative realization of the hydrofacies distribution.*

The validation procedure ensures that different parts of the study area are represented by dividing the boreholes into depth ranges, as used here, or alternatively into zones, borehole clusters, etc., based on their spatial distribution and site characteristics. An independent set of boreholes or a split of the borehole data into two subsets (for model development and validation) can be used, with random selection to reduce bias. Validation showed that R8 was the most plausible hydrofacies distribution among the 10 generated stochastic realizations in the studied aquifer. Despite the use of a coarse grid resolution

in the MA1 simulations, the prediction accuracy remains within an acceptable range, comparable to previous studies in similar heterogeneous settings. In addition, the MA2 simulation analysis revealed that integration of soft data, i.e., ERT-derived hydrofacies lens lengths, provides a slight improvement in model prediction accuracy compared to models based on borehole data alone. To understand the balance between model performance and computational efficiency, model prediction accuracy was analyzed as a function of grid resolution. The results show that the optimal cell size is the one that closely matches the

lengths of the hydrofacies lenses. High-resolution grids failed to improve predictions despite capturing finer details, while coarser grids provide a simplified hydrofacies representation that may improve model prediction accuracy, but at the expense of the spatial resolution of the hydrofacies representation. At first impression, if the performance differences between models using different grids are small, it may be preferable to choose a coarser grid to reduce computational requirements. However, the obvious disadvantage is the potential loss of the ability to resolve specific geological features of interest, which could limit

their use in developing reliable hydrofacies models. Therefore, future research efforts should focus on using hydrofacies models developed with different grid resolutions and evaluating their reliability through numerical groundwater flow simulations.

**Author contributions.** IK and MJ designed the methodological workflow. TM contributed to the conceptualization and

provided access to the input data for model development. All authors participated in the field ERT measurements. EPG interpreted the ERT data and designed the synthetic models. IK developed the T-PROGS models and performed the

simulations. MJ performed the validation and supervised the work. IK prepared the original draft, with contributions from all co-authors, followed by review and editing.

**Competing interests.** The authors declare that they have no conflict of interest.

**Acknowledgements.** The authors would like to thank the Varaždin Utility Company (VARKOM) for providing input data for the model development, and Maja Briški for assistance with field measurements. While preparing this work, the author(s) used AI tools for minor language editing. After using this service, the authors reviewed and edited the text as needed and take full responsibility for the content of the publication.

**Financial support.** The presented research was conducted in the scope of the internal research project „NITROVERT" at the Croatian Geological Survey, funded by the National Recovery and Resilience Plan 2021–2026 of the European Union – NextGenerationEU, and monitored by the Ministry of Science, Education and Youth of the Republic of Croatia. It was also supported by the Croatian Scientific Foundation (HRZZ) through the Mobility Program – outgoing mobility of senior research assistants (MOBODL-2023-08-4470) and by the Slovenian Research Agency (research core funding Groundwaters and Geochemistry (P1-0020)).

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
