# Peer review of "Integrated approach for characterizing aquifer heterogeneity in alluvial plains"

_EGUsphere, 2025_

## Author Response (AR1)

Subject: Final author comments (ACs)

Title: Integrated approach for characterizing aquifer heterogeneity in alluvial plains

Author(s): Igor Karlović et al.

MS No.: egusphere-2025-327

MS type: Research article

We sincerely thank the reviewers for their thorough evaluation and valuable suggestions to improve our manuscript. Below, we provide point-by-point responses to all comments (reviewer's text underlined), followed by the corresponding revisions made to the text. We begin by responding to the comments from Reviewer 1 (Thomas Hermans, RC1), followed by Reviewer 2 (Anonymous Referee, RC2), and finally the community comments from Lee Slater (CC1).

**RC1 (https://doi.org/10.5194/egusphere-2025-327-RC1)**

Dear authors,

I read with interests this paper entitled: "Integrated approach for characterizing aquifer heterogeneity in alluvial plains". In this article, a methodology is proposed to integrate geophysical data as a constraint for geostatistical simulations meant for generating realistic realizations of alluvial aquifer heterogeneity. The topic is relevant, as these aquifers are amongst the most complex to characterize, while there are often exploited for drinking water production and highly contaminated in and around cities due to industrial activity. I support any effort related to a better characterization of these complex systems. In that sense, the paper is interesting, but from my point of view, the reader is left with a feeling of unfulfilled expectations. My major concerns are described below:

1. Geophysical data are not really used as a soft constraint in this study, but merely to estimate correlation lengths. This is a strong limitation of this work since correlation lengths from tomographic methods are known to be biased (Day-Lewis and Lane, 2004). This likely

explains the limited added value of ERT for the evaluation criteria compared to other studies that used ERT in a similar context (e.g. Hermans et al., 2015). Actually, looking at your objectives, I see many similarities withy the study by Hermans et al. (2015). They used ERT to constrain hydrofacies simulated by MPS. They also used falsification to deduce the most realistic training image. They further constrained their simulations by hydrogeological data collected during a pumping test. Also see the work by Barfod et al. (2018) for a similar approach. Therefore, I think it is important to better describe the global context of using geophysical data to constrain geological models, and discuss the proposed methodology in that perspective. Ideally, it would be good to also include simulations where the geophysical data are actually used as soft constraints for the detailed area where data is available.

We acknowledge that geophysical data are not used as a "soft constraint" in exact same context as in the works cited by the reviewer. In our study, these data serve to estimate the length and geometry of characteristic sediments with distinct hydrogeological properties, i.e. hydrofacies units. As Turner (2021) explains, soft data consist of "indirect observations of geological properties, as well as qualitative and interpretative information from geophysical surveys or conceptualizations of the depositional system." We maintain that our approach using ERT data still represents soft data, though applied in a different context. To ensure clarity for readers, we have modified the text to provide a better description of our methodology and the broader context of using geophysical data to constrain geological models, as suggested by the reviewer (also in line with another reviewer, CC1). Throughout our revised paper, we have consistently accompanied the term "soft data" with its specific description as "ERT-derived lens lengths." We emphasize that our ERT-derived soft constraints (both vertical and lateral) are grounded in the actual vertical positions of hydrofacies units observed in wells, addressing the reviewer's concerns about potential bias. We also note that this does not contradict or is in conflict with the work of Day-Lewis and Lane (2004).

Regarding the limited added value of ERT data, the observed similarities between ERT-derived lens lengths and T-PROGS default values in Model Area 2 offer a potential explanation for the similar prediction outcomes. We outline this interpretation in Section 3.2.2 and our response to CC1.

Based on the clarifications made about the methodology and general use of geophysical data, we consider unnecessary to perform MPS simulations to estimate the facies probability linked to a dc-resistivity value (or range of values) as suggested by the reviewer (similar to cited works). However, we recognize the value of this perspective and will certainly consider applying such methods for estimating key hydrogeological parameters in future research.

Reference: Turner, K. A.: Discretization and Stochastic Modeling, in: Applied Multidimensional Geological Modeling: Informing sustainable human interactions with the shallow subsurface, edited by: Turner, K. A., Kessler, H., and van der Meulen, M. J., John Wiley & Sons, Ltd, https://doi.org/10.1002/9781119163091.ch13, 2021.

2. The introduction should focus more on studies which investigated heterogeneity characterization, which is also the topic of this paper. Here are a few references that are relevant, there are many more (and more recent): Baines et al. 2002; Bowling et al. 2005, 2007; Bersezio, Giudici, and Mele 2007; Mastrocicco et al. 2010; Doetsch et al. 2010, 2012a. Gottshalk et al., 2017.

We have carefully reviewed all references suggested by the reviewer and have incorporated those most relevant to support our research.

3. The methodology uses thresholds on resistivity to classify the deposits in hydrofacies (L120-122). But with such a low number of validation wells, the uncertainty cannot be captured, does it? See Hermans and Irving (2017) for a study dedicated to uncertainty analysis in a similar context. This paper shows that such a threshold actually does not exist. For any resistivity value, every hydrofacies has a specific probability of occurrence. This comes mostly from the limitation of geophysical inversion which smooths interfaces, but can also results from the heterogeneity within the sediments. This has also been demonstrated by Isunza-Manrique et al. (2023) in another context. To me, this aspect is essential for any study aiming at a robust integration of geophysical data in a stochastic framework. I would also extract 1D resistivity distribution at the location of borehole and show in parallel the hydrofacies (L182). Because of the smoothing effect of inversion, I really doubt that fixed boundaries can be used. It could reveal a lack of co-located data to derive these ranges.

We recognize that the number of wells used to define the resistivity thresholds is limited to two wells located near four ERT profiles. However, the resistivity anomalies (defining hydrofacies units up to 20 m depth in each profile) are very well defined, with clear resistivity contrasts between them that align with vertical lithological contacts observed in the wells. This result is very important to consider, because it allows us to confidently establish resistivity thresholds for each hydrofacies within the first 20 m depth, without uncertainty. Thus, we differ with reviewer's point of view of implicitly suggesting that uncertainty analysis would show each hydrofacies having a probability of occurrence for any resistivity value, implying no distinct resistivity thresholds exist. From a mathematical perspective, this approach appears unsustainable given the physical and chemical properties of porous media under an electric field. For instance, between clays and gravel, resistivity thresholds can be clearly defined based on well-understood physical characteristics (grain size, pore geometry, grain density, tortuosity) and chemical properties (mineral composition, CEC), along with saturation conditions and pore fluid salinity. The clear differentiation of materials in the Varaždin ERT data (through resistivity thresholds) arises from their distinct physico-chemical properties, producing strong resistivity contrasts. This interpretation is validated by its consistency with vertical lithological contacts observed in wells.

Regarding the limitations of geophysical inversion using the R2 code (ResIPy), we have used synthetic modeling to address potential artifacts associated with the geometries and resistivity contrasts among hydrofacies units in the upper part (above 20 m depth), as well as for the conductive materials at 20 m depth or below (Sgcs, CSs) with unknown geometries (as explained in the text). We also used it to address the limited depth of investigation caused by conductive materials at greater depths (Sgcs, CSs), along with the typical loss of resolution with increasing depth. The smoothed interfaces observed in all synthetic models were not associated with artifacts when using reasonable theoretical (true) resistivity values. We obtained very similar results to those for the field data, validating the quality of field measurements and inversion protocols. Also, synthetic modeling suggests that each unit in the field is in its interior not heterogeneous (at least for the resolution and sensitivity of used electrode array), and that the smoothed interface in the inverted model are reliable. Overall, these synthetic modeling results demonstrate that the smoothness-based regularization approach (Occam's inversion) used by R2 (ResIPy) does not produce any misinterpretation

linked to the smooth appearance of electrical resistivity distribution. In fact, it effectively resolves the geometric characteristics of hydrofacies with minimal residual error, meaning it fits the synthetic resistivity data very well. Therefore, for the field data-based model, we conclude the inversion settings used in R2 were the most suitable (relative to a blocky inversion approach for example) for characterizing the expected smooth-shape structures of alluvial deposits at the study site, resulting in an excellent fit, as indicated by the low misfit error.

While the study by Isunza-Manrique et al. (2023) provides valuable insights, its focus on artificial environments such as municipal landfills makes it fundamentally incompatible with our investigation of natural deposits. These artificial settings involve complexities that extend far beyond simple heterogeneities, which is why we consider their methodology inappropriate as a reference for our work.

In our revised manuscript, we have clarified that our approach does not involve robust integration of geophysical data within a stochastic framework, but is rather related to estimating lens dimensions. In our response to the first comment, we have explained why determining probability of occurrence for each unit is unnecessary in our study context. Following the reviewer's suggestion, Fig. 4 displays the resistivity values extracted from the ERT profile (1D resistivity distribution) alongside borehole SPV-5. To avoid confusion and maintain clarity, there is no need to show the boundaries of $\rho_{hf}$ values or resistivity lines contours next to the borehole (SPV-5) that correspond to different lithologies in Fig. 4, as suggested by RC1. To address this point, we have provided a detailed explanation in the text describing how these values were determined (Section 3.1), which are also shown above the resistivity color scale in Fig. 4 for direct comparison. The fixed resistivity boundaries for each unit using a smooth resistivity distribution are completely validated from synthetic modeling results. Given the type of deposits in the alluvial system and the horizontal dimensions of hydrofacies of several tens of meter, we would expect to find rounded units with flat surfaces as those in synthetic models, which can be perfectly delineated with Wenner or Wenner-Schlumberger array configurations, as we have successfully achieved in our work.

4. I find n=10 realizations a very low number to look at uncertainties. In particular, the objective of a stochastic studies should be to deduce posterior probabilities, and not only deduce the most likely model (see line 162). I think more realizations are needed given that 4 facies are considered.

   During hydrofacies model development, four alternative modeling approaches were tested alongside the maximum entropy method (Carle, 1999), each with 10 stochastic realizations. The results showed no statistical deviation from the maximum entropy method's performance range. Since these alternatives offered no substantial advantage while increasing computational demands, they were excluded from further analysis. Given this, we proceeded with the maximum entropy approach for subsequent analysis, as the method provides intuitive way to observe vertical transition patterns between hydrofacies.

   While we acknowledge that additional realizations could benefit uncertainty quantification studies, this work focused primarily on methodological development for subsurface heterogeneity characterization and identification of the most probable hydrofacies model, as it forms the basis for defining hydraulic conductivity fields. Future work will expand on utilizing the hydrofacies model as direct input for groundwater flow and contaminant transport simulations to support groundwater management in the study area.

5. Some important elements of the methodology are unclear. For example, it is not explained how the correlation lengths are extracted from the geophysical inversion (L169-170). It is crucial to the methodology, and any subjective element or involved parameters should be identified, also considering my other remarks (see point 3 above).

   In the revised manuscript, we have provided a detailed step-by-step explanation of our approach for estimating lens dimensions in Sections 2.2 and 3.1. Briefly, we used a joint interpretation of ERT data, borehole records, and synthetic ERT modeling to systematically characterize all identifiable hydrofacies lenses and estimate their lengths across the 10 ERT profiles. Based on these measurements, which are now presented in the updated Table 1, we calculated mean length values for each hydrofacies type.

6. L197-203. This part is presented as one of the novelty of the paper. However, as explained above, other studies using TI for hydrofacies simulations proposed a more thorough analysis of this (see Hermans et al., 2015 for MPS, and Hermans and Irving, 2017 for synthetic studies and Gottshalk et al. 2017 for indicator simulations). Here, a deterministic approach is first used to derive some correlation lengths that are then integrated in geostatistical simulations, this is not really an integration of ERT into stochastic modelling, which would imply some consideration of the related uncertainty (i.e. probability distributions). Note also that the approach of using synthetic models to validate interpretation is not new. See Caterina et al. 2013 for example. The modelled lenses are very thin. Can they really explain the low resistivity observed, or is it simply a loss of resolution with depth ? How can you be sure only the background profile is found at depth?

We revised the sentence related to the novelty of the approach used in our work, since we did not perform full integration of ERT into stochastic modelling (i.e. probability distributions). Regarding synthetic modeling, we have revised the text to remove any implication of novelty, as the reviewer correctly pointed out that this technique is not new. While we do not cite the work of Caterina et al. (2013), the exercise of carrying out synthetic modeling dates back to the early days of the computer era, initially applied to VES, or to modeling lateral changes and near-surface inhomogeneities prior to ERT. However, we would like to emphasize that, based on the synthetic modelling results, the conductive anomaly at 20–22 m depth plays a key role in defining the depth of investigation for ERT measurements in the Varaždin area, and consequently affects the resolution at greater depths. These two aspects are clearly and easily observed in both the pseudo-3D model shown in Fig.3 and the refined ERT interpretation of each measured line, such as VIN-1 shown in Fig. 4. We are aware, and fully agree with RC2, that synthetic models are equally affected by the conductive layer (CSs-Sgcs lenses or layer), i.e., simulated ERT measurements will exhibit loss of resolution and limited depth of investigation at around 20 m depth. As Reviewer 2 (RC2) points out, our synthetic models explicitly incorporate prior geological knowledge, which cannot be neglected in synthetic modelling. Borehole data confirms the presence of a conductive layer formed by CSs and Sgcs materials with a thickness of approximately 3 m at this depth. The extent of low resistivity values below the depth of the CSs-Sgcs material (whether as a thin layer or lenses) is explained as an artifact due to the loss of ERT resolution, limited by current

penetration depth, rather than representing a true geological feature. Furthermore, borehole data reveal that beneath the conductive CSs/Sgcs layer lies the Gsc hydrofacies, which should exhibit significantly higher resistivity relative to CSs/Sgcs. Refined deterministic analysis of ERT data from iso-resistivity contours maps was very effective in confirming that the apparent extensive conductive zone below 22 m depth in VIN-1 is caused by two distinct conductive anomalies whose depths are underestimated by the inversion approach (i.e., their iso-resistivity contour lines are observed deeper, about 35–40 m). This is observed in both the synthetic model and field data, demonstrating that the CSs-Sgcs materials are highly conductive and significantly affect the ERT data (resolution and depth of investigation). Overall, our expectation from synthetic modeling was to define a plausible geometry of the upper boundary of these materials (as a thin layer or lenses), then constrain their thickness based on borehole data for final interpretation and estimate horizontal lengths used in statistical modeling. This exercise forms the foundation of synthetic modeling work and represents best practice in any geophysical investigation.

Specific comments:

We have carefully addressed all of the specific comments to the best of our ability. For those suggestions that did not include specific line references, we have made reasonable interpretations about their intended locations in the manuscript. Should any of these require adjustment, we would greatly appreciate further clarification to ensure we implement the suggested improvements precisely.

1. There are too many technical details for an abstract. It is not understandable what the grid refers to (ERT inversion grid, hydrofacies simulation grid?). I also miss some context. Typically an abstract should be structured as: 1) Global context 2) Specific research gap 3) Proposed methodology 4) Main results 5) Conclusion.

   We have restructured the abstract to follow the recommended format, avoiding excessive technical details while providing clearer context and research approach. We believe that the revised version more effectively communicates our findings to a broader audience.

2. The resolution of ERT always decreases with depth.

We fully agree with the reviewer's observation about ERT resolution decreasing with depth. While our results show the characteristic resistivity decrease with depth (Fig. 4), borehole data demonstrate that coarser, more resistive units underlie the fine-grained Sgcs-CSs hydrofacies lenses. This highlights that thin conductive lenses mask deeper resistive formations. The current tends to flow through conductive materials, which fundamentally limits depth of investigation and affects resolution (i.e. its capability to demarcate the boundaries of such conductive materials). The text has been revised to clarify that borehole data support this interpretation.

3. Not clear what is meant by boundary conditions in this context.

The text has been revised to eliminate potential confusion.

4. MPS can also be pixel-based. The original SNESIM algorithm was a pixel-based MPS algorithm (Strebelle, 2002), so is the direct sampling algorithm (MAriethoz et al., 2010).

We appreciate the insight and have revised the text to improve clarity, avoiding overgeneralization of simulation methods.

5. "Depending on" rather than "Given".

We have implemented the suggested modification in Section 2.1.

6. Reference to specific Excel functions is not necessary.

We have implemented the suggested change by removing the Excel function reference in Section 2.3.1.

7. Reference to an automated python script is not necessary. It is expected that you made the process automatic.

All Python references have been removed from the validation methodology sections for both model areas (2.3.1 and 2.3.2) as suggested.

8. A resistivity of 4600 Ohm.m seems very high for alluvial sediments. Is this realistic?

The 4,600 ohm·m reflects an inversion artifact resulting from using 10,000 ohm·m as an upper boundary constraint during ERT inversion. Synthetic modeling demonstrates that hydrofacies G exhibits resistivities of 500-1,500 ohm·m, which aligns with expected values for coarse alluvial sediments (as seen in Fig. 4).

9. I don't see any sharp boundaries in the figure, which is a result of the smoothness constrained used for inversion. Have you considered other inversion approaches (blocky inversion, minimum gradient support, etc.)?

We have revised the text to describe boundaries as gradual transitions rather than sharp contrasts, consistent with our use of smoothness-constrained inversion. This method is more adequate to represent the deposition environment, reflecting the expected smoothed, rounded geometries of different materials in the study area.

10. Figure 4. You select different values for the different lenses, why? Why is the CSs layer discontinuous? Wouldn't a continuous layer also explain the data?

Our approach integrates borehole data, ERT measurements, and synthetic model simulations. The geometries of the materials and their resistivity values in these models were constrained by ranges from the field ERT data, as outlined in Section 3.1. A fundamental step in synthetic modeling involves evaluating different initial (true) resistivity values within the possible range for geologic materials, which are primarily influenced by moisture content and, consequently, salinity variations. It is well-known and standard practice to correlate bulk resistivity with electrolyte (pore water) resistivity as a function of moisture and/or salinity variations at laboratory scale. However, in the Varaždin study area, field measurements surprisingly revealed minimal variation, i.e., each material or hydrofacies exhibited well-defined, nearly constant resistivity values, with only slight differences depending on their spatial position (see Fig. 3 for example). Based on this observation, it is reasonable to infer that moisture and/or salinity variations may occur within the same material at different locations. Accordingly, we assigned slightly different true resistivity values to represent natural variability. For the Sgcs-CSs materials, we used values of 150 and 180 ohm·m near the surface and 20 ohm·m at greater depths, reflecting the expected decrease in resistivity with depth due to increasing moisture content and salinity. For the G materials, we applied

true values of 1500 ohm·m and 1000 ohm·m, with the 500 ohm·m difference representing plausible spatial variations in moisture or salinity.

Revisiting RC2's remark, prior geological knowledge constrains our synthetic modeling. While a continuous layer CSs could also explain the data, borehole records clearly demonstrate its discontinuous nature across the study area, as several boreholes did not intersect this layer. Additionally, the resistivity contrast between Gsc and CSs-Sgcs in the first 0-3 m depth is clear, showing the discontinuity (i.e., lenses) of these materials. Note again that the moisture content between these two types of textures plays an important role in generating such resistivity contrasts. We have clarified these points in the revised manuscript to improve understanding.

11. What is "the virtual position of the borehole"? Do you mean a projection of the borehole on the profile?

Thank you for this observation. We have revised the text to use more precise terminology in the Section 3.1.

12. Table 1 gives mean values and ranges, but it is not mentioned how many lenses are detected to calculate them.

As suggested, we have added the number of lenses (n) used to calculate the mean lengths in Table 1.

13. Figure 8. The number of correct predictions is quite low. What would be the score if the most abundant facies (background?) would be predicted everywhere?

The results in Figure 8 are based on 1,200 stochastic realizations, showing a range of correct predictions, with mean values used to compare performance across different grid resolutions. While the number of correct predictions may seem low, the key focus was to evaluate the relative improvement in predictive capability between models developed using only borehole data and those incorporating ERT-derived lens lengths (in addition to testing grid resolutions). The choice of Gcs as the background hydrofacies is the most logical from a sedimentological point of view, representing transitional deposits between coarser and finer

hydrofacies and reflecting shifts between high- and low-energy depositional environments. This interpretation is supported by the ERT profile in Fig. 4, which shows Gcs located between other hydrofacies lenses. Predicting the most abundant facies everywhere would oversimplify the system, ignoring the spatial heterogeneity our method aims to resolve. Such an approach would produce accuracy scores matching the hydrofacies proportions in borehole data, but fail to capture realistic geological structures. As our goal is to reproduce plausible heterogeneity, this hypothetical scenario is not directly applicable. However, if assessed, the accuracy would align with the relative hydrofacies proportions observed in boreholes, as outlined in Section 3.2.2.

14. It is not clear to me what "oversegmentation of the lenses" is. Aren't you overinterpreting distributions that are not significantly different given the low number of samples (n=10)? Wouldn't you need many more validation data to analyze the risk of oversegmentation?

We evaluated grid resolution effects using 1,200 simulations in Model Area 2. The results demonstrated that optimal cell size corresponds to estimated lens lengths. Increasing model detail beyond these dimensions did not improve accuracy, making finer grids computationally inefficient. We acknowledge that "oversegmentation" was an imprecise term and have modified it throughout the text.

15. L335-340. Maybe refer to the work of Danish colleagues who developed a scale of "reliability" when building their geological models using hard and soft data (e.g., Enemark et al., 2024 and references therein).

The suggested paper, while valuable, addresses propagating interpretation uncertainties from 3D hydrostratigraphic models to groundwater models, representing a different research phase from our current borehole-based hydrofacies identification in section 4 (1). We maintain our qualitative reliability framework (defined in Fig. 1) as the appropriate approach for this phase of research, though we will consider the suggested methodology in future research.

16. L342-359. See my main comments related to other studies which proposed more advanced methodologies.

We reviewed the suggested literature and adopted relevant elements that we believe added value to our manuscript.

**RC2 (https://doi.org/10.5194/egusphere-2025-327-RC2)**

General comments

Thank you for submitting this manuscript. Personally, I am curious about applying stochastic modelling for the purposes of electrical resistivity response of the subsurface, so I found it an interesting and engaging read. The reasoning, or wider context, behind the research is well formed (in that the aquifer which is the subject matter of the research is critical to the local water supply). The methodology in this manuscript seems competent, though I'm not familiar with the T-PROGS software that underpins the stochastic modelling efforts. I find the results insightful, however, I do have concerns about the statistical significance the stochastic outputs with and without the benefit of geoelectrical information.

Regardless I wish the authors well, and hope my comments are useful to them. Furthermore, I would like to recommend that this manuscript be accepted on the basis of moderate revisions.

Specific comments

Lines 93 – 95: Is it known how K is determined in these boreholes? Are they determined in situ (e.g. falling/rising head tests) or via laboratory investigations on samples.

The reliability classes presented in Figure 1 were developed to reflect the range and quality of available hydrogeological data in the boreholes, where parts related to information for K determinations refer to all possible estimation methods, including grain-size distribution analyses, permeameter tests from borehole samples, field investigations such as slug tests, pumping tests etc.

Lines 198 – 202: As the ERT is sensitive to 20 m below the ground level, I agree that the synthetic modelling should also be limited to 20 m. Although, I'm unsure if the inclusion of synthetic modelling improves the sensitivity of the electrical measurements to various hydrofacies below 20

_m depth. If one proposes a tomography model (be it through a smoothness regularised gauss newton approach, as with R2, or via stochastic approaches) the raw data is still fundamentally only sensitive to the upper 20 m in this case. It's a limitation of where the electrical current flows, as the authors point out (Lines 198-199). Nevertheless, I think the authors could argue here that the synthetic modelling benefits ground model development by incorporating prior knowledge of the geology._

We agree with the reviewer that both field data-based and synthetic models are reliable down to a depth of approximately 20 meters. More precisely, this sensitivity limit corresponds to the depth where the conductive CSs/Sgcs layer is encountered, e.g. in borehole SPV-5 these two materials form thin conductive layer ranging from 20.7 to 23 m depth. We fully agree with the reviewer's observation that ERT measurements are fundamentally sensitive to the upper 20 meters, regardless of the approach used to estimate a possible geological model. This is the primary reason why Model Area 2 was limited to the upper 20 meters, allowing us to assess how ERT-derived lens lengths and varying grid resolutions influenced model prediction accuracy, while avoiding potential uncertainties arising from synthetic modeling at deeper levels. The purpose of employing synthetic modeling was not to improve sensitivity of ERT measurements, but rather to assess or test different possible geometries, improving the ERT imaging interpretation. The synthetic modeling was beneficial because it provided reliable and realistic hydrofacies dimension estimates that were used as critical input parameters for the T-PROGS model, which in turn improved the procedure for constructing the hydrofacies model in the subsurface. We acknowledge that our original text may have created some confusion about ERT sensitivity limitation, and we have carefully corrected this in the revised manuscript in Section 2.2.

_Figure 4: How did the authors build the lens shapes in the mesh? The results are quite exciting. What is the data misfit of the traditional tomography section and synthetic model versus the real data?_

In the revised manuscript, we include a step-by-step description of our methodology for estimating lens dimensions in Sections 2.2 and 3.1. The synthetic model development in ResIPy was informed by both ERT field data interpretations and borehole records. Particularly valuable were the borehole data, which provided information about depth intervals and thicknesses of individual

hydrofacies units. Some boreholes did not intersect the CSs and Sgcs layer, indicating its discontinuous, lens-shaped nature. In the revised manuscript, we included the final RMS misfit values for both the synthetic model and real data inversions in the caption of Fig. 4.

Figure 5 b: Referring to Figure 2 (which shows a 3D visualisation of the boreholes) the "Clay to silt, sandy" (CSs) hydrofacies appears to dominate the near surface, furthermore, previous studies (Karlović et al., 2021) suggest that the geology is layered. Yet in this figure (5 b) the CSs hydrofacies distribution has little lateral continuity. Can the authors comment on why that might be?

The main purpose of developing the hydrofacies model was to more accurately characterize subsurface heterogeneity. Previous aquifer characterizations relied on simplified, layer-based conceptual models that often overlooked the covering layer. As discussed in the text, this layer is typically thin or absent, suggesting high infiltration potential and increased groundwater vulnerability to surface contamination. Consequently, the layer likely exhibits limited lateral continuity near the surface. Fig. 5b was constructed using 80% of the boreholes from Fig. 2 as hard constraints, ensuring the presence of the CSs hydrofacies at borehole locations. Another important aspect is the model scale, the entire Model area 1 domain (27,984 m × 16,142 m × 100 m) was discretized into 1,000,000 cells (100 × 100 × 100 in the x, y, and z directions), resulting in a horizontal resolution of approximately 280 m × 161 m per cell.

Line 254: What is not clear to me is which information from the electrical resistivity tomography is included or how it is included into the T-PROGS software. Is it just the lens lengths as stated on line 167?

This observation aligns with Reviewer 1's comments (RC1). In response, we have revised the text to more clearly define the role of ERT data as soft constraints within our methodological framework, specifically as ERT-derived hydrofacies lens lengths.

Figure 7 b: See above comment about Figure 5 b.

Similar to our response regarding Fig. 5b, the lateral continuity of the CSs hydrofacies is limited at the surface of Model Area 2 (Fig. 7b), but remains well-constrained at borehole locations. Note that the hydrofacies model in this figure has a horizontal cell resolution of 20 m × 20 m. Moreover,

Fig. 3 effectively illustrates this pattern, as high-resistivity materials, consistent with the gravel and sand observed during field ERT measurements, are frequently present at the surface of the Vinokovščak wellfield.

Lines 308 – 309: Is an improvement of 0.3 to 5.0 % statistically significant enough to argue that the inclusion of ERT has improved the outcome of stochastic modelling? My experience with Markov chain Monte Carlo methods is that the answer can differ for repeated runs by a few percentage points. If the T-PROGS software is utilising Markov chains as the authors state (Line 136) then the question is whether the results are repeatable.

Both approaches (combining borehole data with either ERT-derived lens lengths or default lens lengths) produce a range of prediction accuracies. The mean values for all tested horizontal grid resolutions ($10 \times 10$ m, $20 \times 20$ m, $40 \times 40$ m, $60 \times 60$ m, $80 \times 80$ m, and $100 \times 100$ m) consistently demonstrate better performance with ERT-derived lengths across 1,200 realizations (Fig. 8). While these results demonstrate repeatability, the observed improvement remains slight, as noted in the Abstract, Sections 3.2.2 and 4 (4). Other reviewers (RC1 and CC1) similarly observed the modest improvement in the prediction of hydrofacies distribution when ERT is added. In our discussion in Section 3.2.2, we propose a possible explanation to comparable results, related to similarities between ERT-derived lens lengths and T-PROGS default values (set at $10 \times$ hydrofacies thickness). Additional details are presented in our response to CC1.

**CC1 (https://doi.org/10.5194/egusphere-2025-327-CC1)**

This is an interesting contribution to studies that aim to improve hydrofacies distribution by combining electrical geophysical datasets with discrete borehole logging datasets. The approach is applied to an important study area and addresses issues with nitrate contamination of critical aquifers used for water supply.

The comments posted to date identify some key areas for improvement. In particular, I agree with the need to [1] better define this contribution within the scope of previous similar (foundational?) work [2] better recognize the limitations of using smooth electrical resistivity tomography (ERT) inversions for defining discrete hydrofacies boundaries, and [3] reassess the use of the synthetic

models to assess the reliability of the estimated hydrofacies models. With respect to the third point, I am not entirely sure what was done. The approach described by the following statement is hard to follow: "This issue has been addressed by introducing synthetic models into the ERT interpretation, based on the assumption that if the ERT imaging from synthetic modeling matches or closely resembles to ERT imaging obtained from field measurements, the interpretations and resulting hydrofacies models are reliable". As already noted, it is not reasonable to synthetically infer structures at depth beyond the resolution of the acquired field measurements. Perhaps I am misunderstanding the strategy applied. I suggest that the revisions include a better illustration/justification of the approach with a concrete example of the value added.

In response to point [1], we have reviewed the literature and incorporated relevant prior research, including studies on heterogeneity characterization and previous investigations that utilized geophysical data to constrain geological models.

We have addressed point [2] in our response to comment 3 by RC1. Although we acknowledge that the regularization parameter helps determine a stable solution, we must note that it can also introduce spurious heterogeneity in the solution. However, as demonstrated in our synthetic modeling, the Occam's inversion approach employed by R2 is very effective for determining the regularization parameter to achieve the smoothest possible model that fits the data within acceptable misfit errors, while avoiding both over- and under-fitting.

Regarding point [3], we recognize that the original text may have caused some confusion. We used synthetic modeling as a well-established and reliable tool to assess the sensitivity of our field electrode arrays, depth of investigation, and capability to electrically resolve lithological contrast and geometry between different subsurface units. Beyond these applications, the results of synthetic modeling were very useful to strengthen and validate the final hydrofacies model developed for the Varaždin study area. We have revised the manuscript to describe how synthetic modeling contributed to different aspects of our analysis. Below, we summarize the specific purposes and advantages of our synthetic modeling approach:

1) To validate the reliability of field ERT measurements to depths of 20 m with a very high quality/noise ratio. More precisely, this depth corresponds to the appearance of conductive CSs/Sgcs layer. This was confirmed by synthetic models, resolving the lithology and geometry

within this depth range, suggesting a range of resistivity values consistent with those from field measurements.

2) To detect and validate the influence of the Sgcs and CSs hydrofacies at 20-23 m depth. With synthetic models, we were able to estimate the conductivity/resistivity magnitude of these materials, confirming their role in limiting the depth of investigation of our field ERT measurements. Synthetic modeling results were very useful, demonstrating that the Sgcs/CSs materials exhibit high conductivity at these depths, greater than initially anticipated. This explains the conductive anomaly visible in our ERT imaging (Figs. 3 and 4).

3) To determine whether the Sgcs/CSs materials at 20-23 m depth form lens-shaped features or constitute a single layer. The Sgcs/CSs units are not at depths beyond resolution, but are rather responsible to limit the depth of investigation and resolution. This was the main challenge concerning geophysical analysis. Using synthetic models, we wanted to evaluate the effects of these two possible scenarios (lens-shaped and single layer) to establish the most probable geometry and resistivity magnitudes of these materials. The lens-shaped geometry shown in Fig. 4 emerged as one plausible configuration. Our acceptance criteria for this interpretation rely on strong similarities in the shape of the conductive anomaly appearing at 20 m depth in the synthetic model and the equivalent anomaly in the field-data inversion model. We note that the thickness of the layer is consistent with well-constrained measurements from borehole data. Moreover, since this layer is absent in certain boreholes, we infer that it is not a continuous across the study area.

We acknowledge that the original text lacked clarity regarding this analysis. In response, we have substantially revised Sections 2.2 and 3.1 to provide a better explanation of our strategy applied and justification for the modeling approach. With these textual improvements clarifying the purposes and results of our approach, we believe the illustrated example in Fig. 4 now appropriately demonstrates our findings.

One additional point to consider in the revisions is that the paper implicitly assumes that variations in grain size primarily control variations in electrical resistivity. Of course, resistivity also depends strongly on porosity, fluid conductivity, and the degree of saturation. Although variations in porosity sensed with resistivity could contribute to improved hydrofacies discrimination, variations in fluid conductivity and degree of saturation would likely complicate the delineation

of hydrofacies. Is there any field data available to constrain the magnitude of these variations? Presumably, water tables are available in the wells. Significant variations in fluid conductivity might be expected given the nitrate contamination problem. Perhaps these complicating factors can partly explain why the improvement in the prediction of hydrofacies distribution is modest when ERT is added? Some further discussion of this issue is required.

This paper will make a valuable contribution to the hydrogeophysics literature once the comments provided in the posted discussions are addressed.

While we acknowledge that electrical resistivity depends on multiple factors including porosity, fluid conductivity, and degree of saturation, our study considers grain size variations as the primary control for hydrofacies discrimination. Although variations in fluid conductivity and saturation could potentially complicate facies delineation, our observations in Model Area 2 demonstrate limited saturation effects for hydrofacies G. Long-term water table records (2005–2019) from observation wells (prefix SPV; Fig. 1) reveal water table fluctuating between 3.68 and 10.95 m, depending on the hydrological season, wellfield pumping regime, proximity to pumping wells, and Drava River levels. More recent field measurements from monthly campaigns in the Vinokovščak wellfield area show a stable water table, ranging from 4.49 m to 7.65 m (median: 6.74 m). Field measurements from 2018-2022 revealed stable groundwater electrolytic conductivity (479-501 μS/cm, median = 492 μS/cm), indicating consistent total dissolved solids content. Due to its negative charge, nitrate is mobile, resulting in conservative transport behavior. We expect that nitrate mobility follows the hydrofacies sequence: G > Gsc > Sgcs > CSs. In the Varaždin aquifer, nitrate concentrations are typically higher in shallow zones due to proximity to contamination sources. Unfortunately, multi-depth screened wells at the Vinokovščak wellfield yield mixed samples during pumping, preventing depth-specific parameter analysis and any direct correlations with resistivity measurements. As shown in the Fig. 4, the high-resistivity lens associated with hydrofacies G at the beginning of the profile exhibits comparable resistivity values in both dry (surface) and saturated (deeper) zones. Variations between lenses along the profile (500 ohm·m in the field data model, and 700 ohm·m in the synthetic model computed using average resistivity values) suggest these differences are primarily spatial rather than related to saturation state.

Degree of saturation in the vadose zone is expected to be a function of lithology, i.e. it decreases with increasing the grain size and pore radius. At shallow depths, the resistivity of Sgcs-CSs material exhibits resistivity values twice as high as those at 20-25 m depth in the saturated zone (low resistivity, 60 ohm·m from inverted field data, 140 ohm·m estimated from synthetic model and 20 ohm·m suggested as the true resistivity value from synthetic model). The hydraulic conductivity (K) of the four hydrofacies units is the subject of a separate paper currently under review. While these results are not yet published and we cannot discuss them in detail, our methodology involves estimating K values from grain-size distribution curves using empirical methods, followed by validation against pumping test data. The derived K ranges for individual hydrofacies show clear separation, with only minor overlaps occurring in transitional deposits (Sgcs and Gsc) that vary in their sand and gravel ratios, as extensively documented in the literature.

However, our analysis revealed that similarities between ERT-derived lens lengths and default T-PROGS values (set at $10 \times$ hydrofacies thickness) may account for the comparable prediction outcomes between the two approaches, a point we have now included in our discussion in Section 3.2.2. The ratios of ERT-derived to default lens lengths vary across different hydrofacies, with hydrofacies G showing ratios of 2.9–3.1, Sgcs 1.8, and CSs 1.2–1.5 (derived from the data in Table 1). While hydrofacies G exhibits the greatest dimensional discrepancy, its volumetric predominance within the system compensates for these variations, making it the dominant control on the overall hydrofacies model.

---

## Author Response (AR2)

Subject: Final author comments (ACs) – round 2

Title: Integrated approach for characterizing aquifer heterogeneity in alluvial plains

Author(s): Igor Karlović et al.

MS No.: egusphere-2025-327

MS type: Research article

We appreciate the Reviewers' careful evaluation and suggestions, which have helped us improve our manuscript. Below, we present point-by-point responses to each comment (with the Reviewer's text underlined), along with the corresponding revisions made to the manuscript. We begin by responding to the comments from Reviewer 1 (Thomas Hermans), followed by Reviewer 2 (Anonymous Referee).

**Referee #1: Hermans, Thomas**

This is my second review of your paper. Although this version is largely improved, I have the feeling you misinterpreted some of my comments so that they are not entirely integrated in the new version. Through the clarifications you made, I understand this is not so crucial, as you only use the ERT data to derive estimated correlation length, but it is important that everything written is correct and that the discussion is clarified, since reviewer comments are also accessible along the published article.

I would therefore like to come back on some of my previous comments and your answers to them.

1. You are not using the ERT-data as a direct constraint to the facies. I would however classify approaches which did it as more advanced compared to what you do. I therefore think it is important this is highlighted more in the conclusion, and certainly as a perspective of the work. Such approach would make a true difference in the estimation of the facies (see 10.5194/hess-22-3351-2018, 10.5194/hess-22-5485-2018, examples with TEM data, but basically similar, and Hermans et al. (2015), cited but without a thorough discussion). This is really crucial to put your work in perspective of the existing literature.

We have revised the main text (in the Introduction and Summary and conclusions sections) to better contextualize our approach, as suggested by the Reviewer. We have clarified that our study employs the classical inversion approach or method (also referred to as deterministic or local inversion method), rather than a probabilistic approach based on stochastic regularization. Although there are some examples highlighting the advantages of stochastic regularization scheme (several modalities) in solving inversion problems, this approach remains an alternative, and not a replacement for the classical and widely preferred deterministic inversion method (e.g., Ramirez et al., 2005). The Reviewer argues that the alternative inversion method based on a stochastic regularization might yield better final model results (e.g., such approach would make a true difference in the estimation of the facies) as demonstrated in Hermans and Irving (2017). However, in our study, the ERT measurements resolve hydrofacies geometry identified by boreholes quite well (except below 20 m depth, as discussed in the text). Moreover, we correlate the real depths of different hydrofacies observed in boreholes with the iso-resistivity values that are consistent with

the type of material, i.e., consistent with the electrical properties of each material. Given that the classical inversion method based on smoothness regularization constraint produces models that are geologically plausible and consistent with the lithological reality observed in the boreholes, we find it well justified for our case. That said, we agree that stochastic and deterministic methods can be complementary, as noted by Ramirez et al. (2005). Even in Hermans and Irving (2017), both approaches resolve the inversion problem similarly (their Figure 9a, 9b).

Ramirez, A. L., Nitao, J. J., Hanley, W. G., Aines, R. D., Glaser, R. E., Sengupta, S. K., Dyer, K. M., Hickling, T. L., and Daily, W. D.: Stochastic inversion of electrical resistivity changes using a Markov Chain Monte Carlo approach, J. Geophys. Res., 110, 1-18, https://doi.org/10.1029/2004JB003449, 2005.

2. You write: However, the resistivity anomalies (defining hydrofacies units up to 20 m depth in each profile) are very well defined, with clear resistivity contrasts between them that align with vertical lithological contacts observed in the wells. This result is very important to consider, because it allows us to confidently establish resistivity thresholds for each hydrofacies within the first 20 m depth, without uncertainty.

Writing "without uncertainty" is a very strong and, I am afraid, wrong statement. You should remember that any estimation you made is based on an inversion that distorts the true resistivity value. In addition, your hydrofacies are observed in boreholes that are not co-located. As a result, you don't know where the interfaces are actually lying on your profile. It is impossible to state that it lies at a specific contourline. Even if you knew the exact location of the interface, It would likely not lie at the same contourline everywhere, because of the limitation of the inversion. This is actually confirmed as your interpretation involves different value for the transition between facies. It means that there is some subjectivity in the selection of these contourlines.

We agree that phrase "without uncertainty" may overstate it, and "with less uncertainty" would be more accurate. If we want to be rigorous, all geophysical methods are inherently subjective to some degree. In this study, and ERT investigations in general, it is impossible to know the real resistivity distribution of the subsurface. Instead, we must rely on the physics and chemistry governing electrical properties of geologic materials in order to make reasonable interpretations on the most plausible model provided by the data inversion, while ensuring consistency with geologic reality (that again, we will never know it using geophysical approaches). The concerns raised by the Reviewer are valid and were already considered in our analysis. Precisely because

inversion is an imperfect process that distorts the "true resistivity," we grounded our interpretations in the physics behind the electrical properties of the materials. This approach allows us to correlate inverted resistivity values (yielded by an imperfect data inversion process) with the most appropriate geologic materials.

In our study, ERT results were interpreted through analysis of material electrical properties. The high-resistivity anomalies observed in Fig. 4 are associated gravel (G) hydrofacies. Although SPV-5 isn't co-located with the ERT profile, the thickness of high-resistivity anomalies matches the G unit observed in SPV-5, a discrepancy we attribute to lens-shaped depositional geometry. Similarly, conductive CSs-Sgcs materials at shallow depths and at about 20m depth show thicknesses consistent with those observed in SPV-5. In both cases, the resistivities of G and CSs-Sgcs materials (their boundaries) can be delineated by the contour resistivity lines in the ERT imaging as we have done, reducing the uncertainty linked to the limitations of the inversion process by correlating the contour resistivity lines with the thickness (vertical contacts) of materials observed in the borehole SPV-5. The different values for the transitions between facies (gradual resistivity changes) is what defines the intrinsic subjectivity of the resistivity method, which cannot be avoided. The interpreter's experience and scientific knowledge play a crucial role in developing realistic ERT interpretations. In this study, the reliability of our results is supported by three key factors: high-quality field measurements, proper application inversion approach (regularization criteria), and a strong consistency between vertical distribution of the subsurface resistivity (yielded by the inversion approach) and different materials observed in the boreholes. This alignment, which is consistent with the physics behind the electrical properties of materials, is what makes our interpretation reliable with less uncertainty. As a result, it was possible to estimate the lateral extension of different hydrofacies and incorporate these findings into the T-PROGS model.

3. You write. Thus, we differ with reviewer's point of view of implicitly suggesting that uncertainty analysis would show each hydrofacies having a probability of occurrence for any resistivity value, implying no distinct resistivity thresholds exist. From a mathematical perspective, this approach appears unsustainable given the physical and chemical properties of porous media under an electric field. For instance, between clays and gravel, resistivity thresholds can be clearly defined based on well-understood physical characteristics (grain size, pore geometry, grain

density, tortuosity) and chemical properties (mineral composition, CEC), along with saturation conditions and pore fluid salinity.

This statement is wrong as demonstrated by Hermans and Irving (2017), I really encourage you to read that paper in details. Indeed, you might be able to discrimante a low conductivity clay from a resistive gravel, but your geological settings also has intermediate sand facies. Your error comes from the fact that you are confusing true resistivity values with inverted resistivity values. I invite you to plot the inverted resistivity versus the true resistivity for your synthetic case (Figure 4), and you would likely observe an overlap of inverted resistivity between most facies. This all comes from the limitation of regularized inversion. Eventually, it might not make a big difference for the estimation of hydrofacies characteristic length, but it should be clear that any length you estimate from ERT can be impacted by the inversion process. Indeed, the countour lines you use certainly do not correspond to the real interfaces. And this is the case even if if the choice of your inversion parameters (smoothness constraint inversion with R2) is properly done. What is needed to understand this is present in Figure 4 and can be easily included in the text.

We respectfully note that the Reviewer's response may not fully consider the context of our approach, focusing on points that don't directly engage with our evidence-based interpretation. We used a scientific criterium to interpret the resistivity measurements based on the governing physico-chemical parameters (petrophysical and hydrogeological properties for each geologic material) that define their electrical properties (see our previous response above). After reading both Hermans and Irving (2017) and Hermans et al. (2015), we observed a very clear similarity of both deterministic (smoothness regularization) and stochastic inversion approach (Figure 9a,b in Hermans and Irving, 2017) consistent with lithology shown in Figure 3 of Hermans et al. (2015) – grain size increases gradually with depth from 0 to 10 m, so a gradual increase in resistivity with depth is expected, as it is shown by the deterministic inversion method (their Figure 9a in Hermans and Irving, 2017). Moreover, there are some puzzling aspects in the studies referenced by the Reviewer that warrant discussion. For example, the reported resistivity values, where gravel appears less resistive than sand, and clay lenses in the bottom part of the deposits tend to display values close to those of the gravel facies (Hermans and Irving, 2017). Such findings would require laboratory validation to rule out potential factors like presence of fine-grained particles in coarser facies, decreasing the "true" values of resistivity. Without this physical verification, it's difficult to assess whether their approach offers real advantages over conventional methods. In contrast, in

our study we have classified hidrofacies in detail (see Table 1 for very well classified texture of geologic materials observed in the boreholes), and the ERT images of inverted resistivity are consistent with such textural classes. Even in the mixtures like the Sgcs facies (transitional between clay, CSs and gravel, G-Gsc), the dominant fraction governs the electrical signature. In our study, we are confident (as explained above in the previous response) that in the inverted data (i.e., based on the smoothness regularization constraint of inversion method) we are clearly distinguishing and delimiting all determined hidrofacies, based on realistic electrical properties of the different geologic materials observed in boreholes.

4. You write: In fact, it effectively resolves the geometric characteristics of hydrofacies with minimal residual error, meaning it fits the synthetic resistivity data very well.

You should be careful with what you do with the synthetic case. The only thing you can do is confirm that the proposed geometry in the synthetic case can lead to a similar resistivity distribution as observed after inversion. It does not mean that this interpretation is actually correct. There are plenty of other possibilities that could lead to similar results. This is the definition of a non-unique solution.

We fully agree with the Reviewer's perspective regarding the synthetic model, as it represents just one of many possible subsurface resistivity distributions. While acknowledging this limitation, the synthetic modeling results suggest that our field data inversion model is one of the most likely to occur. Therefore, we have used this criterion for validating the field data inversion process and the final inverted model from which we deduced the distribution of hydrofacies model based on resistivity values, i.e., the vertical and lateral extension of each hydrofacies as we mentioned in the text.

I have some suggestions to further improve the manuscript.

1. L64. "Unregulated" or "excessive" instead of "irresponsible". The latter implies decision despite the knowledge it could be harmful, while I assume it might not be the case.

We appreciate the insight and have implemented the suggested modification.

2. L76. I would add "as conditioning data" and cite some of the papers where it has been done like Hermans et al. (2015, aready cited) or Barfod et al. (2018, 10.5194/hess-22-3351-2018, 10.5194/hess-22-5485-2018). This is really important to properly guide the reader where more

advanced conditioning has been proposed. I would also come back on this in your summary/conclusion since there is no discussion section.

The text has been revised accordingly.

3. L127-131/ See my comments above on your response. You should highlight that the choice of the contour line bears some subjectivity as you don't know the real interface location (borehole are projected) and that you interpret the gradual transition from high to low resistivity. You cannot be certain about that and it is important you acknowledge it explicitly in the manuscript even if the impact is likely limited in your methodology as you are not using ERT as conditioning data (that can also be underlined in the manuscript).

We address this comment by modifying the text in Section 2.2. As we mentioned in previous responses and explicitly mentioned in the text, we used the two observation wells SPV-5 and SPV-8 nearby ERT profiles VIN-1, VIN-4 and VIN-10 to find the sites where each single hydrofacies matched with an iso-resistivity contour line from the overlapping between the projection of boreholes and the ERT profiles, estimating the resistivity values for each geologic material. For instance, the thickness of hydrofacies CSs-Sgcs close to the ground and the top of CSs-Sgcs lens at 20.7 m depth observed in SPV-5 (shown in Fig. 4), matched quite well with their corresponding $\rho_{hf}$ (contour iso-resistivity value) in the projection of SPV-5 over VIN-1, whereas the thickness of G material observed in SPV-8 and the top of CSs-Sgcs lens at 25.5 m depth matched very well with their corresponding $\rho_{hf}$ values in the projection of SPV-8 over profile VIN-4 (and also VIN-10). Based on this approach, we believe the subjectivity mentioned by the Reviewer is mitigated, given the accuracy of the final inverted model and assuming that the $\rho_{hf}$ values are consistent with the electrical properties of geologic materials forming the hydrofacies. However, acknowledging the intrinsic subjectivity of geophysical methods (measurement errors) and inherent to the solution of inversion problem (modelling errors), we modified the text in Section 2.2 accordingly to address the Reviewer's comment.

4. L132. Space between "are" and "using".

The text has been revised accordingly.

5. L140. Space between "data" and "and".

The text has been revised accordingly.

6. L161. Refer to Table 1 when mentioning proportions.

The text has been revised accordingly.

7. L199-200. If a high resistivity is next to a low resistivity, then you will always see a transition to intermediate, even in the absence of the intermediate facies. This is why relying on a single contour line or limited number of boreholes might be misleading. In figure 4, are you therefore always sure there is Gsc between G and Sgcs? I would not be.

What we gather from this comment is that the Reviewer may have misinterpreted the lithology of our study site, and consequently, the ERT results. In Fig. 4, we present one plausible distribution of hydrofacies in the subsurface, based on resistivity distribution. This model, along with our interpretation from the original paper, clearly shows that G and Sgcs occur as lenses embedded within Gsc, which is consistent with the information from boreholes. Near the surface, particularly within the upper 6 meters, we clearly show lateral transitions between Gsc and Sgcs-CSs, or between G and Gsc. These lateral alternations are very well defined by iso-resistivity contours lines.

8. L201. Replace "strong continuity" by "good/satisfactory/acceptable consistency".

The text has been revised accordingly.

9. L212. I presume that the 0.1 is in the log scale and not the resistivity scale in ohm.m.

We corrected the text accordingly. Effectively is in log10 scale. We change 0.1 (log10) to 1.26 ohm·m to be consistent with the resistivity scale in Fig. 4.

10. L212. A kriging interpretation requires a variogram model, that should then be mentioned. I assume however a simple linear interpolation would do the same job (this is basically a downscaling approach).

In this approach, kriging is used to generate the contour lines over the grid of "true" resistivity values yielded by the inversion process. Although kriging interpolates between two consecutive values, this interpolation does not influence the results (though we have taken care to verify this). The iso-resistivity contour lines are purely for visualization, however, they prove highly useful in defining threshold resistivity values for hydrofacies.

The Reviewer's comment extends beyond the scope of our methodological approach, adding additional complexity. For example, Hermans and Irving (2017) and Hermans et al. (2015) display inversion results using contour lines, a standard method for resistivity mapping, without addressing variograms or downscaling techniques. This suggests that the comment may not fully align with the context of our study.

11. L214-216. It is still not fully clear to me. As I understand, the boreholes are projected onto the profiles, there is quite some uncertainty about these transitions. It is ok for your purpose that you define some threshold, but these should not be overinterpreted, and the amount of subjectivity involved should be acknowledged. As mentioned before, I really doubt that you have the exact same isocontour in all boreholes, even if you had co-located data. If it was the case, this would really be a coincidence that would not be validated if you had more boreholes available. Stating otherwise would be in strong contradiction with the existing literature. This is actually implicitly acknowledged in line 200-222 since you have different values of the contour line.

Important to note, our approach advances the determination of hydrofacies lens lengths, parameters typically derived from sedimentary process analogs or literature values when direct data are limited. As explained above, there is no subjectivity in the 2D ERT imaging based on contour lines or iso-resistivity contours. While it is true that the boreholes are projected onto the ERT lines (we intentionally avoided placing them directly on the lines because the metallic casing of boreholes could affect the ERT measurements), the data in Fig. 4, though not co-located, still allow for meaningful interpretation. Using physical principles of electrical properties of geologic materials, and the very clear contrasts of resistivity, corresponding to different materials or hydrofacies, we were able to match the lithology contacts in boreholes with the iso-resistivity lines or resistivity contours not exactly in the borehole projection but few meters laterally. This allowed us to assign reasonable resistivity values to each hydrofacies, in line with theoretical values of resistivity for each type of materials.

12. L234-236. It would be nice to show some of these tests, so that the reader can have a better feeling of the methodology.

There is no need to show some of the tests, because of the non-uniqueness there is an infinite number of models that can provide the same results. The most robust approach is to base the interpretation on the physics behind the electrical signatures measured in the field (resistances,

apparent resistivities) and then on the distribution of resistivities provided by the inversion process ("true" resistivities = best possible resistivity values), then evaluate whether they are true or not based on the known geological and hydrogeological environment. In this study, the real vertical distribution of lithology is provided by the boreholes, and the strong correlation with the ERT images based on iso-resistivity contours lines demonstrate the accuracy of the methodology. The proposed lens-shaped geometry of CSs-Sgcs in the subsurface, initially inferred from boreholes, is clearly resolved in ERT imaging and corroborated by synthetic modeling. Both datasets show the incomplete presence of this CSs-Sgcs layer indicating its discontinuous nature across the study area, consistent with typical alluvial depositional environments.

13. L247-248. This is also linked to their thicknesses.

It might be. However, if Reviewer bases his comments on the transversal resistivity and associated problem of equivalences (Maillet, 1947), we should consider that the model parameters in 2D ERT imaging are the resistivities allocated in pixels at pseudo-depths and not thickness as in 1D (VES). Strictly speaking, the thickness as a model parameter is not considered in 2D inversion problem. Therefore, although physically it is correct to make an assumption that resistivity is affected by the thickness variation of a particular layer, we prefer to interpret that the difference in the orders of magnitudes of resistivity models is likely to be mainly caused by the different intrinsic modeling error levels between the inversion model of field data and that of synthetic models. Furthermore, this analysis is not relevant here, because the objective of the synthetic model was to corroborate the geometry of hydrofacies CSc-Sgcs as lens-shaped, and not to estimate real resistivity values for such material.

Maillet, R.: The fundamental equations of electrical prospecting, Geophysics, 12(4), 529–556, https://doi.org/10.1190/1.1437342, 1947.

14. L255-256. you don't know. You just show that your synthetic model after inversion gives something similar, it does not mean that the true model resistivity is that one. But indeed, the discrimination potential of ERT is high close to the surface (Hermans and Irving, 2017).

Revised version of the sentence excludes ambiguous elements that could confuse readers.

15. L266-267. I am pretty sure one could fine a synthetic model with a continuous lens to represent this. Maybe it would involve variation in resisitivity values like you used for the different gravel

lenses. Synthetic modelling tells you this situation is possible, not that other geometries are impossible.

While alternative interpretations are theoretically possible, the borehole data conclusively demonstrate the discontinuous nature of this layer (as explained in the text and our earlier responses). Our interpretation provides the most likely outcome, consistent with both the observed data and the known characteristics of the depositional system.

16. L290-291 + conclusion (point 1/3). It could also indicate that transition probability is not sufficient to capture the spatial patterns ? Maybe methods which better deal with multiple-point statistics are needed.

Our approach is based on the transition probability method, which has proven effective for evaluating spatial patterns in this study. While methods involving multiple-point statistics may offer additional insights, their application lies beyond the scope of the present work. Nonetheless, we acknowledge their potential and will consider integrating such techniques in future investigations to further strengthen our analysis.17. L309-310. "Two validation boreholes in R8 achieved 100 % prediction accuracy (Fig. 6b), both located in the northwestern part of the study area." It is not relevant, realizations are equiprobable and should not be ranked based on the validation.

The text has been revised accordingly.

18. L329. What does "and any potential ambiguities from synthetic modelling at greater depths" mean? What are the ambiguities from synthetic modelling?

The text has been revised, part of the sentence left out to avoid confusing the reader.

19. L365-369. I fear you are here interpreting variations that are not statistically relevant. Have you run a statistical test?

We interpreted the resolution-dependent patterns and trends emerging from 1,200 model realizations. Statistical tests to assess the significance of differences between sets of realizations were not performed.

20. L385-393. Here, you should mention that more advanced approaches exist to constrain geostatistical models with resistivity, where resistivity values are used as a soft constraint. There

is no doubt such an approach, if resistivity values are broadly available, would result in better identification of facies.

We appreciate this perspective and have addressed this issue in our response to the first comment. In our view, stochastic inversion methods are not more advanced methods than the classical (so-called deterministic) inversion method, but rather serve as valuable complementary approaches to the latter and vice versa (Ramirez et al., 2005). We believe that combining different methodologies is one of the best practices to yield the most robust hydrofacies characterization in heterogeneous alluvial environments. In our study, ERT measurements were sufficiently accurate to provide reliable distribution of the subsurface resistivity, consistent with the lithology and their vertical extent observed in boreholes. Consequently, for our specific research objectives, the ERT imaging methodology (with its limitations), yielded results of sufficient practical value.

21. L401-402. I don't know what robust means in this sentence... ERT is just use to adapt the length of the facies, so there is not reason why TPROGS would fail taking this into account.

The text has been revised accordingly.

22. L408-409. Sentence about R8 is not relevant, see above, especially with such a low number of realizations.

We have removed the sentence, as suggested.

23. L416-417. Suggesting a coarse grid which exceeds the size of identified geometry is strange. I would not do that, especially given the small differences in obtained distributions. In addition, eventually, the size would rather be linked to the requirement of groundwater modelling.

We appreciate this insight and have revised the text to clarify that grid selection must prioritize geological accuracy and modeling needs.

**Anonymous Referee #2**

Thank you for responding to all of my comments from the previous submission. I think any sources of confusion have been addressed. Aims are well formed and the relationship between the electrical resistivity tomography results and the stochastic modelling has been discussed in a clearer manner.

A minor note, on line 390 the authors state that ground penetrating radar (GPR) could be used to aid in hydrofacies delineation in future studies. However the context of the paragraph discusses a highly conductive layer at ~20m depth. GPR's depth of penetration is inversely proportional to ground conductivity, so I doubt GPR would effective in such an environment; even then, GPR traces can be hard to interpret for features several meters below the ground surface (unless on a glacier) and would require a low frequency system (10s of MHz). Seismic methods might be more appropriate than GPR as one would anticipate variations in wave speed between the gravel and clay layers. A variety of EM methods could also be good candidates for studying gravel lenses.

In any case, I'm happy to recommend that the paper be accepted (pending any technical corrections).

We fully agree with the Reviewer's comment and greatly appreciate the suggestion to incorporate seismic methods. Effectively, GPR signals (electromagnetic waves) attenuate in electrically conductive material or under high moisture conditions (water saturation). Our idea for proposing GPR (surface-deployed) was precisely to detect these attenuation patterns, particularly for identifying CSc-Sgcs lenses near the surface and at depths below 20 m. Another advantage of GPR is its significantly higher resolution compared to ERT, so we would expect more detailed delineation of features such as the geometry of G lenses. However, GPR should be very relevant if deployed in cross-borehole or well-to-well configurations, along with cross-borehole ERT and well-logging tools. Following the Reviewer's suggestion, we acknowledge seismic and EM methods would be worth implementing in the study area. In future work, we will try to test some of these methodologies, along with stochastic-based approaches for data processing and inversion.

Specific comments:

There is typo on line 83, should read "of ERT".

Thank you, we have revised the text accordingly.